# A therapeutic regimen using neoantigen-specific TCR-T cells for HLA-A*2402-positive solid tumors

Yuncheng Bei [1,4], Ying Huang[2,4], Nandie Wu[1,4], Yishan Li[1,3], Ruihan Xu[1], Baorui Liu [1✉] & Rutian Li [1,3✉]

## Abstract

The adoptive transfer of TCR-T cells specific to neoantigens preferentially exhibits potent cytotoxicity to tumor cells and has shown promising efficacy in various preclinical human cancers. In this study, we first identified a functional TCR, Tcr-1, which selectively recognized the SYT-SSX fusion neoantigen shared by most synovial sarcomas. Engineered T-cell expressing Tcr-1 (Tcr-T1) demonstrated HLA-A*2402-restricted, antigen-specific antitumoral efficacy against synovial sarcoma cells, both in vitro and in vivo. Furthermore, to extend its application, we developed a cooperative therapeutic modality, in which exogenous SYT-SSX fusion neoantigen was loaded into stimuli-responsive nanoparticles (NPs) formed by mPEG-PVGLIG-PCL copolymers (Neo-AgNPs) for tumor targeting delivery. As expected, Neo-AgNPs were proven to have great tumor penetration and local release. In situ, the modification was able to direct engineered Tcr-T1 against other HLA-A*2402-positive malignant cancer cell lines with significant antigen-specific cytotoxicity despite their inherent mutation profiles. With these favorable data, our established cooperative therapeutic modality has great potential for further clinical investigation and provides new insight for future TCR-T cell therapy development.

**Keywords** SYT-SSX Fusion Neoantigen; TCR-T Cell Therapy; Stimuli-responsive Nanoparticle; Solid Tumor; Cancer Immunotherapy
**Subject Categories** Cancer; Immunology

## Introduction

Immune cells can specifically eliminate cancer cells through tumor antigen recognition. On this basis, great developments in immunotherapy have been made for clinical treatment of various malignancies. Adoptive T-cell transfer (ACT), particularly T-cell receptor-engineered T-cell (TCR-T) therapy, exhibits promising efficacy holding great promise for cancer immunotherapy. Grafting with T-cell receptors (TCRs) that specifically recognize MART-1$_{27-35}$ retarget autologous T cells against progressive metastatic melanoma,

and objective response rate (ORR) was 12% (Morgan et al, 2006). Adoptive transfer of TCR-T cells targeting NY-ESO-1$_{157-165}$ mediated remarkable tumor regression in patients with metastatic melanoma (ORR: 55%) (Robbins et al, 2015) and synovial cell sarcoma (ORR: 61%) (Robbins et al, 2011b). These impressive clinical benefits demonstrate the feasibility and effectiveness of TCR-T cell therapy against solid tumors. However, there remain tremendous challenges for its further application. To our knowledge, tumor antigens presented by human leukocyte antigen (HLA) molecules restrict TCR-T cell activation. Although tumor-associated antigens (TAAs), including differentiation antigens, overexpressed antigens, cancer-testis antigens, and oncofetal antigens, are widely used as the target for the development of TCR-T cell therapy in various malignancies (Jhunjhunwala et al, 2021), the sufficient safety and effectiveness remain to be well documented due to the deficiency of their specificity (Jhunjhunwala et al, 2021; Lahiri et al, 2023).

Recently, fast development and application of sequencing technology, particularly single-cell multi-omics technologies, have greatly contributed to the identification of neoantigens. Unlike TAAs, neoantigens are generated from non-synonymous somatic mutations, specifically expressed in malignant cells making them ideal targets for TCR-T cell therapy. Indeed, the adoptive transfer of TCR-T cells targeting mutated KRAS$^{G12D}$ elicits objective regression in treating metastatic colorectal cancer (Tran et al, 2016). High-avidity TCRs specific for KRAS$^{Q61K}$ (Peri et al, 2021), KRAS$^{G12V}$ (Veatch et al, 2019), p53$^{R175H}$ (Lo et al, 2019), and other hotspot mutations have also been identified. These preclinical studies reveal that neoantigen-responsive TCR-T cells exhibit the ability to preferentially eliminate tumor cells showing little cytotoxicity against normal cells offering great promise of safety and effectiveness. Unfortunately, the high heterogeneity of tumor cells among patients with the same type of malignancy makes it difficult to develop a universal TCR-T cell, resulting in a major limitation for their clinical application. Thus, identifying a conserved neoantigen is crucial for developing TCR-T cell therapy. To our knowledge, most fusion mutations occur in key transcriptional factors (Zhou et al, 2022), which are responsible for ~20% of cancer morbidities (Wang et al, 2021). Peptides overlapping the breakpoint regions of two different genes tend to be more immunogenic than other neoantigens (point mutation, single nucleotide variants, and insertions and deletions), serving as ideal conserved neoantigens for TCR-T cell therapy (Ma et al, 2014; Wang et al, 2021).

Synovial sarcomas are aggressive sarcomas that account for almost 10% of all soft tissue sarcomas (Ladanyi, 2001). Importantly,

[1]The Comprehensive Cancer Center, Nanjing Drum Tower Hospital, The Affiliated Hospital of Nanjing University Medical School, 210008 Nanjing, China. [2]Department of Oncology, The Affiliated Huai'an Hospital of Xuzhou Medical University and The Second People's Hospital of Huai'an, 223022 Huai'an, China. [3]Clinical Cancer Institute of Nanjing University, 210008 Nanjing, China. [4]These authors contributed equally: Yuncheng Bei, Ying Huang, Nandie Wu. ✉E-mail: baoruiliu@nju.edu.cn; rutianli@nju.edu.cn

nearly all synovial sarcomas bear the translocation t(X;18) (p11.2;q11.2), representing the fusion of SSX (including SSX1, SSX2, or SSX4 isoforms) with SYT (Ladanyi, 2001). Therefore, the antigens derived from SYT-SSX fusion could be recognized as "universal" neoantigens. In this study, we followed the idea that isolating and identifying high-affinity TCRs against shared SYT-SSX fusion neoantigen paves the way for the adoptive transfer of TCR-T cells as an ideal approach for synovial sarcomas. To achieve this goal, we first evaluated the peptide sequences of each isoform of SYT-SSX fusion proteins spanning the breakpoint regions, and the shared sequence was subsequently subjected to NetMHC-pan4.0 for the prediction of neo-peptides. The neoantigen-reactive T cells (NRTs) were stimulated via a modified fast in-vitro stimulation protocol (Pathangey et al, 2017). Subsequently, the neoantigen-specific TCRs were analyzed by comprehensive use of single-cell transcriptome and TCR repertoire profiling and their functionality and specificity toward synovial sarcomas with SYT-SSX fusion mutation were further identified in vitro and in vivo. More importantly, to broaden its application we developed a cooperative therapeutic strategy: a well-defined stimuli-responsive tumor-targeted delivery system was utilized to provide adequate neo-peptide to tumor cells independent of their inherent mutation profiling. Cooperatively adoptive transfer of SYT-SSX fusion mutation-specific Tcr-T1 cells exhibited extensive anti-tumoral activity, establishing an inherent mutation-independent tumor-targeted cooperative TCR-T therapeutic regimen.

# Results

## Identification and validation of immunogenic SYT-SSX fusion peptides

Nealy all synovial sarcomas (over 90%) have been identified with a chromosomal translocation between SYT and SSX genes (Appendix Fig. S1) (Ladanyi et al, 2002), resulting in a fusion mutation of SYT to SSXs gene (including SSX1, SSX2, and SSX4). As shown in Fig. 1A, the peptide (PPQPPQQRPYGYDQIMPKKPAE) was dominantly shared by fusion genes SYT-SSX1, SYT-SSX2, and SYT-SSX4. Thus, to identify the universal immunogenic neo-peptides based on SYT-SSX1/2/4 fusion genes, the shared peptide sequence was selected and analyzed by using the NetMHC 4.0 epitope-HLA prediction algorithm as described before (Lundegaard et al, 2008). The predicted neo-peptides and corresponding HLA alleles are shown in Fig. 1B.

To our knowledge, the right epitope of great specificity and ability to induce robust T-cell responses is crucial for the development of effective engineered T-cell therapies. To evaluate the immunogenicity of the selected neo-peptide candidates, we utilized the fast in-vitro stimulation protocol as described before (Pathangey et al, 2017), in order to induce antigen-specific T-cell activation. Briefly, as shown in Fig. 1C, unfractionated PBMCs were firstly exposed to GM-CSF plus IFN-α 2 A for DC differentiation, and LPS and TLR7/8 ligand R848 (Resiquimod) for DC maturation. Subsequently, candidate neo-peptides were added. Antigen-pulsed DC facilitated T-cell activation (Fig. 1C). Finally, exogenous IL-7 was used only for sustained expansion of CD4+ and CD8+ antigen-responsive T-cells. Then, the cells were harvested and subjected to flow cytometric analysis for detection of T-cell

activation, defined by high expression of CD137 (Seliktar-Ofir et al, 2017). The gating strategy was illustrated in Appendix Fig. S2A. IFN-γ production in supernatant was calculated via CBA assay. Consistent with previous findings (Kawaguchi et al, 2005), T-cells were significantly activated in Pep-4-pulsed PBMC from HLA-A*2402 donors characterized by elevated CD137 expression (Fig. 1D) and enhanced IFN-γ production (Fig. 1E). And Pep-3 was able to slightly induce T-cells activation from HLA-A*1101 donor (Appendix Fig. S2B,C) as predicted. However, no significant T-cell activation was observed in the rest groups (Fig. 1D,E; Appendix Fig. S2D-E). For further confirmation, the gelatinase-linked immunospot (ELISpot), a single-cell sensitive membrane-based assay for immune response detection, was performed as described (Leehan and Koelsch, 2015). Antigen-specific T-cell responses (> 100 spot-forming colonies (SFCs)/$10^6$ PBMCs) were found in HLA-A*1101 PBMC pulsed by Pep-1, or Pep-3, and in HLA-A*2402 PBMC pulsed by Pep-1, or Pep-4 (Fig. 1F). Particularly, Pep-4 induced the highest level of T-cell response (Fig. 1F), possibly because of the strongest binding activity between HLA-A*2402 molecule and Pep-4 defined by peptide affinity assay (Fig. 1G). More importantly, T-cells from HLA-A*2402 PBMC after indicated stimulation exhibited an antigen-specific cytolytic activity against Pep-4-pulsed HLA-A*2402+ target cells (Fig. 1H). Collectively, these data indicated that Pep-4 was a potential HLA-A*2402-restricted T-cell epitope, and of great capacity to induce HLA-A*2402-restrict antigen-specific T-cells activation.

## ScRNA-seq and scTCR-seq identified HLA-A*2402-restricted SYT-SSX-specific TCR repertoire

Next, Pep-4 peptide was applied for further HLA-A*2402-restriced SYT-SSX-specific TCRs identification as shown in Appendix Fig. S3A. Briefly, SYT-SSX-specific T-cells were expanded via two-step fast stimulation protocol as described before, and Pep-4-specific T-cell activation was determined by IFN-γ production and target cell killing assay. Then, CD8 + T-cells were isolated via a magnetic separation system (Appendix Fig. S3B) and were subsequently subjected to 10X genomics platforms for further single-cell sequencing. Single-cell RNA sequencing (ScRNA-seq) and TCR sequencing (TCR-seq) were parallelly performed.

After filtering cells with low quality and doublet clearance, 36 million RNA transcripts were obtained in 490 T-cells identified by CD3 expression (Appendix Fig. S3C), and complementarity-determining region 3 (CDR3) sequences. 92% (453 of 490) T cells containing TCRs with paired α/β chains were obtained (Appendix Fig. S3C) (GSE243535). Using t-distributed stochastic neighbor embedding (t-SNE), the isolated T-cells were separated into 7 Clusters (Fig. 2A) according to differentially expressed gene signatures (Appendix Fig. S3D). As expected, most of the isolated T cells, especially Cluster 1 and Cluster 2, belonged to CD8 + T-cells (Fig. 2B), and very few of them were CD4 + T cells (Appendix Fig. S3E). For further evaluation, we performed Cluster analysis on each of the 7 Clusters. The classic markers of T-cell subtypes (He et al, 2022) indicated the presence of conventional naive, effector, memory, and exhausted T-cell in the 7 Clusters (Fig. 2C). As shown in Fig. 2C, Cluster 1, characterized by high expression of active mitosis markers (including MKI67, TK1, and STMN1), exhibited effector T-cell phenotype. Additionally, traditional T-cell activation-related genes, such as GZMB, IL-7R, IL-2RA, and IFN-γ were

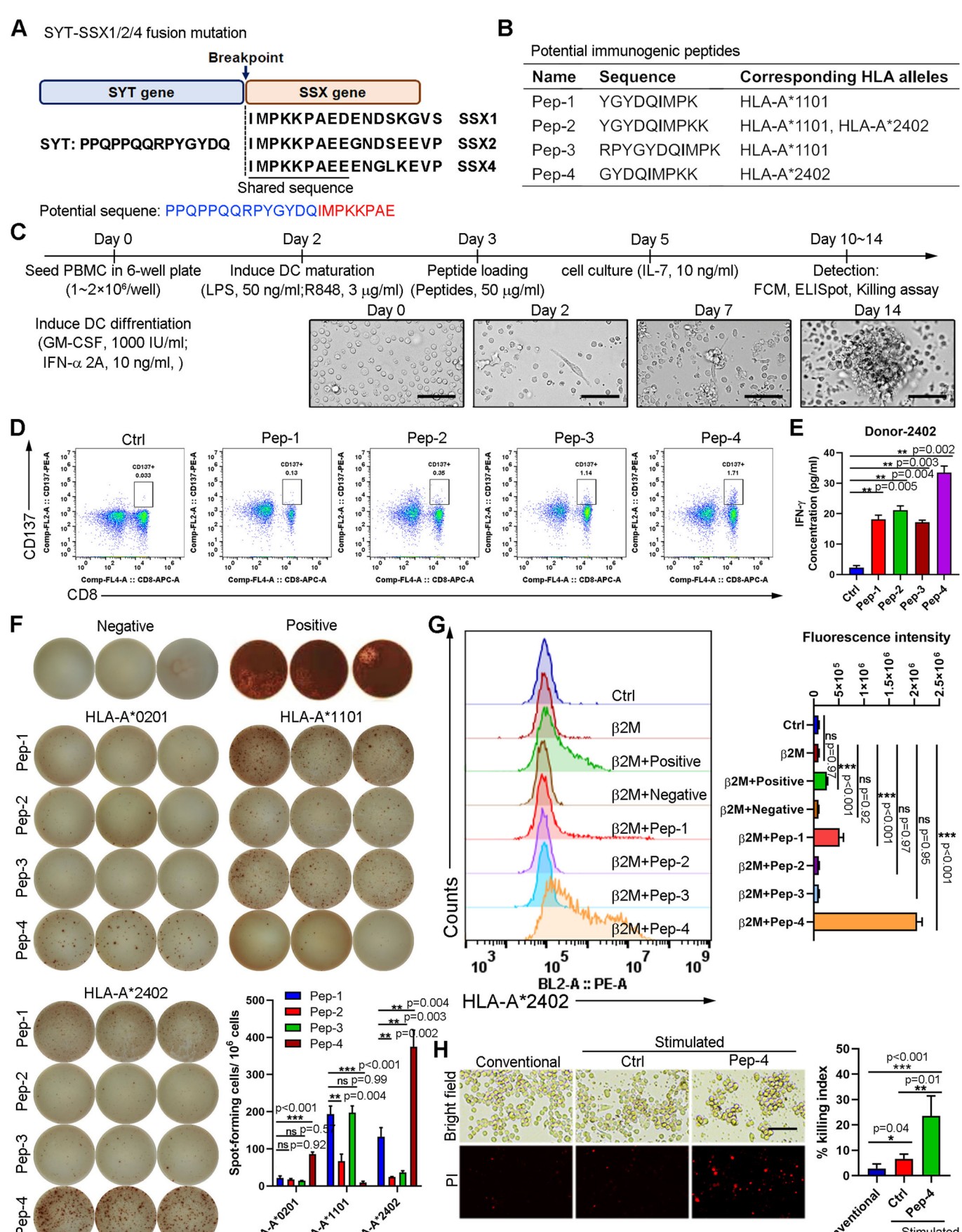

**Figure 1. Fast stimulation of neoantigen-responsive T cells against shared neo-peptide derived from SYT-SSX fusion mutation.**

(A) Schematic diagram of SYT-SSX fusion protein isoforms. The amino acid residues spanning the breakpoint regions are indicated, and the shared sequence is underlined. (B) Potential immunogenic neo-peptides predicted by netMHC-pan4.0 and their corresponding HLA alleles are presented. (C) Schematic illustration of the fast stimulation protocol for induction of neoantigen-responsive T cells from unfractionated PBMC. Microscopic images of antigen-specific T-cell expansion on indicated time during fast stimulation. Scale bar = 50 μm. (D, E) PBMC was isolated from HLA-A*2402 healthy donor and subsequently stimulated via indicated protocol (C). Flow cytometric analysis of T-cell activation marker, CD137, expression in CD8 + T cell subpopulation (D). Levels of secreted IFN-γ after stimulation were measured by CBA assay (n = 3 independent replication) (E). (F) PBMC was isolated from HLA-A*0201, HLA-A*1101, and HLA-A*2402 healthy donor, and stimulated via protocol described in (C) (n = 3 independent replication). IFN-γ ELISPOT assay was performed as described under methods. The images of formed spots in each well were shown. The number of spot-forming cells (SFC) per 10⁶ were summarized. (G) Flow cytometric analysis of the binding activity of indicated neo-peptides against HLA-A*2402 target cells was performed as described in the methods (n = 3 independent replication). Representative flow cytometric images were shown (left). Fluorescence intensity was summarized (right). (H) Pep-4-responsive T cells were stimulated via protocol in (C). Then they were incubated with HLA-A*2402-positive target cell (MKN45) at a ratio of 1:1, with or without Pep-4 pulsing. Unstimulated PBMC was used as a negative control (conventional). After 24 h incubation, dead cells were labeled by PI (n = 3 independent replication). Representative images showed Pep-4-specific cytotoxicity of NRT cells against HLA-A*2402-positive MKN45 cells (left). Scale bar = 50 μm. The killing index (No. of PI-positive cells/No. total cells) was summarized (right). Data are presented as mean ± SEM. Statistical significance was calculated using the one-way ANOVA (E, F–H). ns not significant, *P < 0.05; **P < 0.01; ***P < 0.001. Source data are available online for this figure.

also elevated in Cluster 1 compared with Cluster 2 identified by t-SNE (Appendix Fig. S3F), and volcano plot analysis (Fig. 2D). The GO (gene ontology) analysis strengthened the conclusion that T-cells in Cluster 1 were potentially antigen-responsive (Fig. 2E). These data indicated that T cells in Cluster 1 potentially had experienced antigen-specific stimulation.

Given that antigen-specific T cells were expanded after in-vitro stimulation, the abundance of TCR sequences of responsive T-cells should be theoretically increased. Thus, we analyzed the diversity of the TCR repertoire obtained by sc-TCR sequencing. We noticed that multiple T-cells shared the same TCR sequencing (Fig. 2F), and detailed information was provided in Appendix Table S1. According to t-SNE analysis, we found that Tcr-1, Tcr-2, and Tcr-3 were dominantly enriched in Cluster 1, while others were distributed in different Clusters (Fig. 2G; Appendix Fig. S3G). According to the heatmap shown in Fig. 2H, we noticed that T-cell activation-related genes were highly elevated in Tcr-1-expressing T-cells. Similar results were also observed in Tcr-2-expressing and Tcr-3-expressing T-cells, but not in other subclones of T-cells (Fig. 2H). These data indicated that Tcr-1, Tcr-2, and Tcr-3 exhibited great potential to respond to Pep-4 neo-peptide. Furthermore, differentially expressed gene (DEG) analysis discovered that in Tcr-1-expressing T-cells (identified as an expression of TRAV12-1, TRBV3-1 in Fig. 2I), T-cell cytotoxicity associated genes (GNLY (Guo et al, 2018), ZNF683 (Caielli et al, 2019), GZMB (Loebbermann et al, 2012), CXCL13 (He et al, 2022), and PRF1 (Givechian et al, 2018)), and inflammation related gene (FXYD2) were significantly elevated (Fig. 2J; Appendix Fig. S3H).

Collectively, Tcr-1-expressing T cells exhibited effector T-cell phenotype and were characterized with active mitosis and elevated cytotoxicity. These data indicated that Tcr-1 was of great potential as an HLA-A*2402-restricted SYT-SSX-specific TCR.

## Generation and characterization of SYT-SSX-specific TCR-T cell

Since the possible TCR repertoire of Pep-4-specific HLA-A*2402-restricted T-cells was identified, we planned to generate TCR-T cells and investigate their functional specificity and efficacy. To this end, the piggyBac transposase (PB) transposition system was applied (Ding et al, 2005). As indicated in Appendix Fig. S4A, the codon-optimized human TRA and TRB sequences were linked by the 2A sequence, and the constant regions were replaced by mouse

constant regions to increase correct pairing. A selection and safety marker RQR8 (Philip et al, 2014), a highly compact marker/suicide gene containing CD34 and CD20 epitopes, was co-expressed with TCR by cloning it downstream of the TCR separated by the 2A sequence. EGFP was also cloned downstream of RQR8, which was used for expression detection. Detailed amino acid sequences are presented in Appendix Table S2. The protocol for PB transposition, purification, and expansion of human TCR-T cells is depicted in Fig. 3A. Representative FACS plots and statistics during different stages of TCR production demonstrated that transduction efficacy was nearly 20% and purification resulted in over 90% pure TCR-T population (Fig. 3B).

To determine the peptide specificity and HLA-A*2402 dependency of constructed TCR-T cells, the established TCR-T cells expressing Tcr-1 (Tcr-T1), Tcr-2 (Tcr-T2), and Tcr-3 (Tcr-T3) were respectively incubated with HLA-A*2402-positive target cell (MKN45 labeled by luciferase, MKN45-Luci) with or without Pep-4 pulsing (Fig. 3C). As shown in Appendix Fig. S4B, Tcr-T1 exhibited significant Pep-4-specific activation characterized by elevated IFN-γ production when incubated with Pep-4-loaded HLA-A*2402-positive target cells. Although high production of IFN-γ was observed in both Tcr-T2 and Tcr-T3 treatments compared with conventional untransduced T-cells (UTD), no differences were observed in groups with Pep-4 incubation (Appendix Fig. S4B).

For further functional validation of the Tcr-T1 cells, we perform an in-vitro assay to explore the Pep-4-specific and HLA-dependent cytotoxic activity of T cells against synovial sarcoma cells. To this end, SYT-SSX fusion mutation-negative, HLA-A*2402-negative human synovial sarcoma cell line SW982 was applied and further transgenically modified as previously reported (Matsuda et al, 2018). Briefly, plasmid DNAs designed to express a part of SYT-SSX fusion mutated protein (~50 amino acids of SYT-SSX fusion gene and the mutation was placed in the center), or the same sequence with Pep-4 deletion (SYT-SSX ΔPep-4) (Appendix Fig. S4C). HLA-A*2402 was linked with these mutated proteins via T2A (named SYT-SSX/2402, SYT-SSX ΔPep-4/2402 respectively) and cloned into an expression vector (Appendix Fig. S4C). Downstream EGFP was used for expression examination. Purified target cells SW982 (Appendix Fig. S4D) with indicated transgenic modification were incubated with Tcr-T1 cells, and cellular cytotoxicity was determined via the luciferase assay mentioned in the methods. UTD T cells were used as a negative control. As expected, Tcr-T1 cells, rather than UTD T cells, exhibited cytotoxic

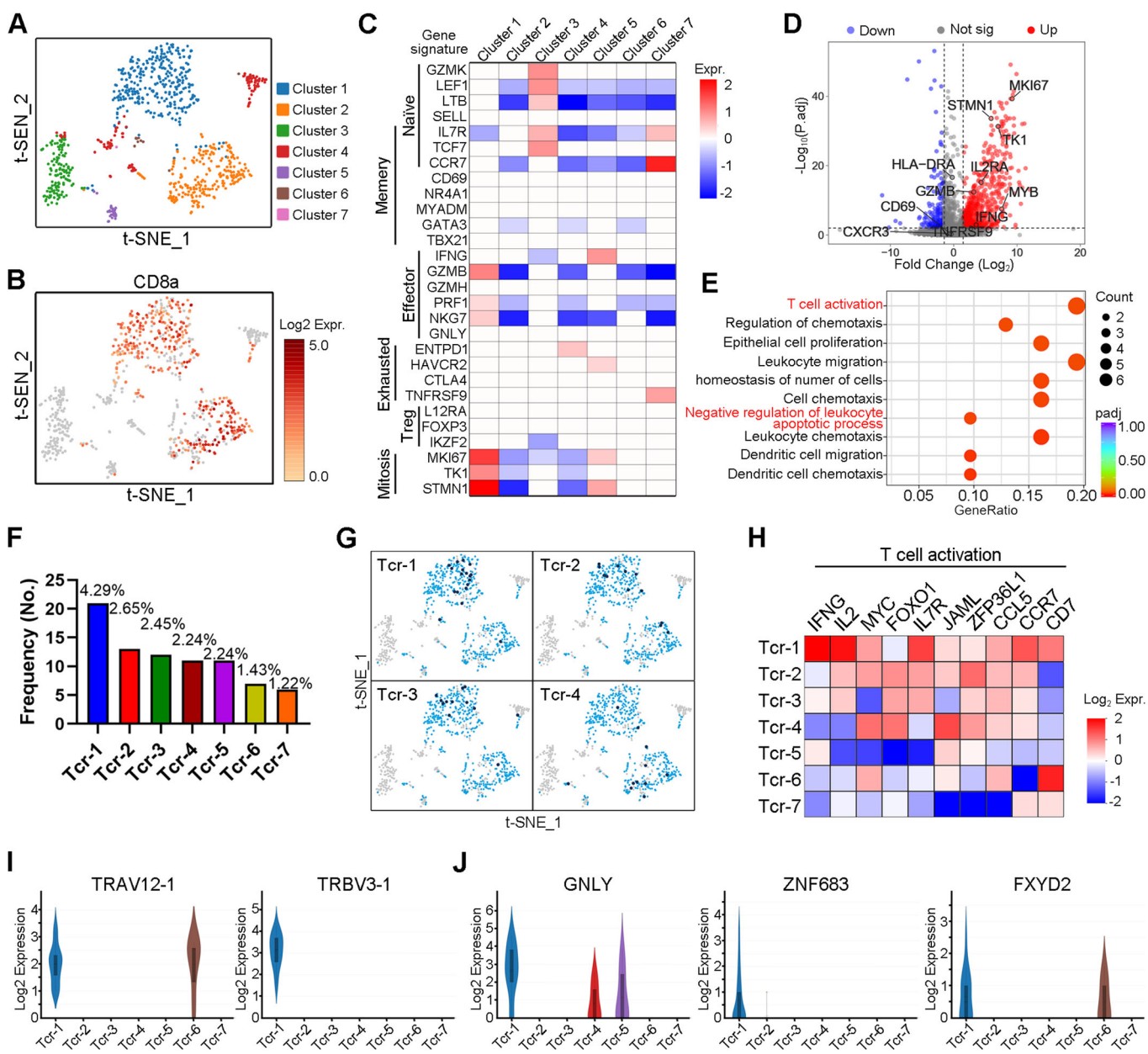

**Figure 2.  scRNA-seq for massively parallel measurement of TCR sequence, and transcriptome.**

(A, B) NRT cells were stimulated through the application of Pep-4 based on PBMC derived from HLA-A*2402 healthy donor. CD8 + T-cell subclone was isolated via magnetic beads and subjected to 10x Genomics for scRNA-seq. Independent tSNE plots of isolated NRT cells (A). Each dot represents a cell; each color indicates a distinct T-cell Cluster (A). CD8 + T cells were highlighted (red) (B). (C) Heatmap showed mean expression of genes associated with T cell subtypes in each Cluster. (D) The volcano plot showed significantly altered genes in Cluster 1 compared with Cluster 2. (E) The GO analysis indicated signal pathway was significantly enriched in Cluster 1 compared with Cluster 2. (F, G) The profile of TCR usage in stimulated NRT cells. The frequency of Top 7 TCR used by stimulated NRT cells (F). tSNE plots of subclonal T cells using indicated TCRs (G). (H) Heatmap showed the T-cell activation-related gene expression in subclonal T cells using each of the top 7 TCRs most frequently used. (I, J) Violin plots of effector T-cell marker genes in each T-cell subclone (Tcr-1, n = 21; Tcr-2, n = 13; Tcr-3, n = 12; Tcr-4, n = 11; Tcr-5, n = 11; Tcr-6, n = 7; Tcr-7, n = 6). Statistical significance was calculated using the Student's t test (D, E). Source data are available online for this figure.

activity exclusively against SYT-SSX/2402 SW982 cells (Fig. 3D; Appendix Fig. S4E). The deficiency of either Pep-4 neoantigen or HLA-A*2402 restricted Tcr-T1-induced cytotoxicity against SW982 (Fig. 3D,E). IFN-γ secretion assay revealed that Pep-4 mediated activation of Tcr-T1, but not UTD T cells in a dose-dependent manner (Fig. 3E). Similar results were observed against

HLA-A*2402-positive gastric cell line MKN45 (Fig. 3F). However, no significant cytotoxicity was observed in T-cells expressing other TCRs (Appendix Fig. S4F). These data indicated that Tcr-T1 showed Pep-4-specific HLA-A*2402-dependent cytotoxicity against synovial sarcoma cells, as well as other HLA-A*2402-positive tumor cells.

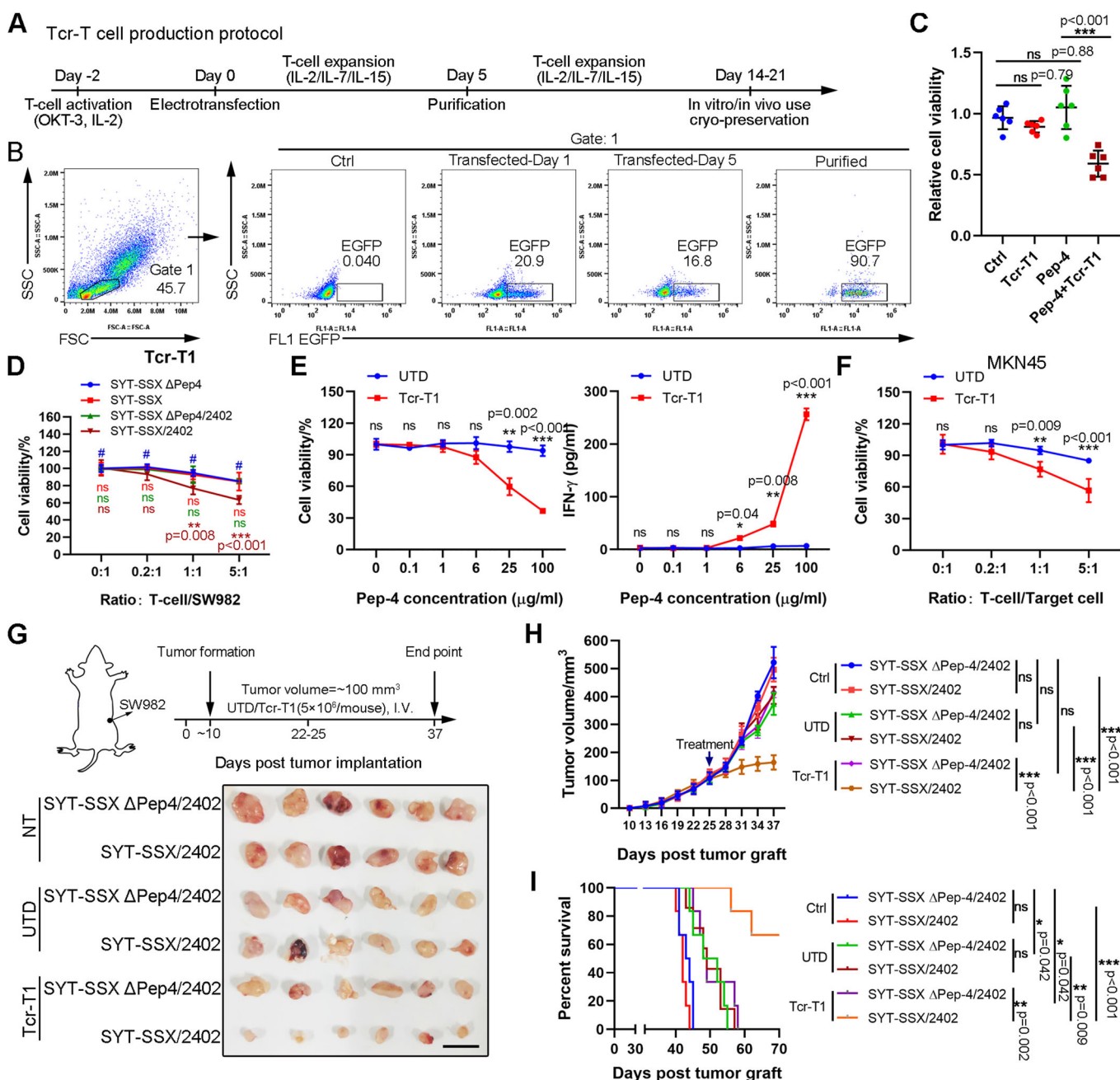

**Figure 3. Tcr-T1 cells exhibited antigen-specific cytotoxicity against HLA-A*2402-positive synovial sarcoma cells in vitro and in vivo.**

(A, B) Tcr-T1 cells were generated as mentioned in the methods. Schematic for TCR-T cell generation, purification, and in vitro expansion. (A) Representative flow cytometric images indicated the electroporation efficiency and purification of Tcr-T1 cells (B). (C) HLA-A*2402-positive MKN45-Luci cells were treated with Tcr-T1 cells, with or without Pep-4 pulsing. 24 h later, cell viability was tested via CCK-8 assay. (D) Human synovial sarcoma cell line SW982 was modified with stable expression of the HLA-A*2402 molecule to generate target cells (SW982/2402). Then these target cells, SW982 and SW982/2402, were transfected with vectors expressing SYT-SSX fusion mutated peptide containing/deleting Pep-4 sequence (SYT-SSX ΔPep4, SYT-SSX, SYT-SSX ΔPep4/2402, and SYT-SSX/2402). Target cells were incubated with Tcr-T1 cells at ratio 0:1, 0.2:1, 1:1, and 5:1 (T-cell/SW982). Cell viability was determined by using CCK-8. (E) SW982 cells overexpressing HLA-A*2402 and SYT-SSX fusion mutated peptide containing Pep-4 sequence (SYT-SSX/2402) were incubated with Tcr-T1 cells (E: T = 1:1). Cell viability was determined by CCK-8. IFN-γ secretion was examined using a CBA assay (n = 3 independent replication). (F) Cytotoxicity of Tcr-T1 against HLA-A*2402-positive MKN45 was determined as indicated (n = 3 independent replication). (G–I) Treatment scheme (G, up). SYT-SSX/2402 and SYT-SSX ΔPep4/2402 cells (1 × 10^7 cells/mouse) were injected subcutaneously in nude mice (n = 6). When tumor volume reached to ~100 mm^3, Tcr-T1 cells, and UTD T cells (5 × 10^6 cells/mouse) were intravenously injected. Mice were sacrificed at the endpoint. Representative photographs of tumor tissues isolated from each group of mice (G, bottom). Tumor volumes of each group were recorded (H). The overall survival of each group was recorded. (I) Data are presented as mean ± SEM. Statistical significance was calculated using the Student's t test (C, E), one-way ANOVA (C), two-way ANOVA (D–F, H), and Kaplan–Meier survival analysis (I). ns not significant, *P < 0.05; **P < 0.01; ***P < 0.001. Source data are available online for this figure.

## Adoptive transfer of Tcr-T1 cells suppressed SW982-formed tumor growth in vivo

To assess the efficacy of the adoptive transfer of Tcr-T1 in vivo, we used a xenograft model in which transplant tumors were established in immunodeficient mice using transgenic SW982 cells as described before (Appendix Fig. S4C). As depicted in Fig. 3G, synovial sarcoma cell line SW982 with different transgenic modifications was subcutaneously implanted into nude mice. When tumor volume reached ~100 mm³, tumor-bearing mice received a tail vein injection of UTD T-cells or engineered Tcr-T1 cells ($5 \times 10^6$ cells/mouse). Treatment of Tcr-T1 cells injection efficiently suppressed the growth of tumors formed by SYT-SSX/2402 SW982 cells, and significantly prolonged survival, compared with a control group receiving UTD T-cells (Fig. 3G–I). However, the lack of either HLA-A*2402 molecule or Pep-4 neoantigen restricted Tcr-T1-induced anti-tumoral activity in vivo, indicating the Pep-4-specific HLA-A*2402-restricted cytotoxicity of Tcr-T1 cells in vivo (Fig. 3G–I). More importantly, no significant weight loss was observed in each group throughout the experiment (Appendix Fig. S5A), indicating no significant systemic toxicity. To further evaluate the side effects of Tcr-T1 therapy, the major organs were subjected to histological analysis with H&E staining to evaluate the toxicities. Consistent with our expectation, no noticeable damage was observed in the spleen, kidney, liver, lung, and heart, as well as in serum biochemical assays (Appendix Fig. S5B,C).

## Immunophenotypic analysis of Tcr-T1 cells

To better understand the anti-tumoral activity of Tcr-T1 cells, we performed flow cytometric analysis to evaluate the immunophenotypic property of Tcr-T1 cells at the end of treatment. To this end, single-cell suspension was prepared as previously described (Bei et al, 2020), and then subjected to flow cytometric analysis. The gating strategy was shown in Appendix Fig. S6A. Consistent with previous finding, CD8 + T-cells, derived from SYT-SSX/2402-formed tumor tissues with Tcr-T1 transfer, were activated characterized by elevated expression of CD137 (Fig. 4A), while little difference of CD137 expression was observed in either CD8+ or CD4 + T-cells from both tumor and spleen tissues derived from other groups (Fig. 4A,B; Appendix Fig. S6B,C).

Given that long-term memory T cells are crucial for engineered T-cell therapy due to their property of inducing more rapid and robust responses upon re-exposure to tumor antigen and facilitating durable anti-tumoral activity (Sallusto et al, 2004), we performed a flow cytometric analysis to verify the alteration of subpopulation of memory T-cell. Effector memory T-cell ($T_{EM}$) was characterized as CD3+CD45RO+CD62L-, which were further divided into CD4+ and CD8+ subpopulations (Appendix Fig. S7A). As shown in Fig. 4C, both CD8+ Tcr-T1 cells and CD4+ Tcr-T1 cells (Appendix Fig. S7B) from SYT-SSX/2402-formed tumors possessed a higher proportion of $T_{EM}$ population compared with T cells from tumors of other groups. Similar results were observed in spleen tissues (Fig. 4D; Appendix Fig. S7C). To our knowledge, programmed death 1 (PD-1) and T-cell Ig and mucin domain-3 protein (TIM-3) are widely reported as two major immune checkpoint receptors highly expressed on activated tumor-infiltrating T lymphocytes (Cai et al, 2023). Indeed, we noticed that PD-1 and TIM-3 were significantly elevated in CD8+ Tcr-T1 cells derived from tumor tissues when exposed to the Pep-4/HLA-A*2402 complex (Fig. 4E,F). However, in CD4 + T-cells, PD-1 was slightly upregulated in CD4+ Tcr-T1 cells derived from SYT-SSX/2402-formed tumors, while there were no differences in TIM-3 expression in CD4 + T-cells among different groups (Appendix Fig. S7D,E). These data not only proved that Tcr-T1 cells were activated dependent on Pep-4 neoantigen expression but also provided hints that combining immune checkpoint inhibitors, especially TIM-3 antibody, might facilitate Tcr-T1 therapy efficacy.

Collectively, these data indicated that Tcr-T1 cells were restrictedly activated upon exposure to Pep-4/HLA-A*2402-positive SW982 cells, and a high proportion of $T_{EM}$ might contribute to durable anti-tumoral activity.

## A cooperative therapeutic strategy for antigen-negative tumors

Given that TCR-T cell reactivity is restricted to antigens presented by tumor HLA molecules, the high heterogeneity of solid tumors makes it difficult to generate universal neoantigens, limiting the clinical application of TCR-T immunotherapy. Even though SYT-SSX fusion mutation happens in almost all patients with synovial sarcoma (Ladanyi et al, 2002), due to the extremely rare incidence of synovial sarcoma (Nielsen et al, 2015), SYT-SSX fusion mutation seldomly happens in other solid tumors. To expand the application of generated Tcr-T1 cells into various types of solid tumors, we developed a TCR-T cooperative therapeutic strategy against solid tumors independent of the inherent neoantigen profile, as shown in Fig. 5A. This therapeutic strategy contained two steps. In the first step, tumor-specific neoantigens were delivered to tumor tissues using a stimuli-responsive nanoparticle. Mechanistically, the stimuli-responsive nanoparticle could enter into tumor tissue via enhanced permeability and retention (EPR) effect, and release multiple cargos in response to certain stimuli (Pilkington et al, 2024). It is widely reported that matrix metalloproteinase (MMP) 2/9 was enriched in various solid tumor tissues, playing a key role in cancer invasion and metastasis (Kessenbrock et al, 2010), which have been regarded as a favorable target for drug delivery (Zandieh et al, 2023). Thus, we generated an MMP2/9-responsive nanoparticle using methoxy poly (ethylene glycol)-polycaprolactone (mPEG-PCL) copolymers as described before (Li et al, 2013). Briefly, an MMP2/9 cleavable peptide, PVGLIG, was inserted between mPEG and PCL blocks. Then, the Pep-4 neo-peptide was loaded into mPEG-PVGLIG-PCL copolymer, and delivered into tumor tissues via EPR effects (Zandieh et al, 2023). Upon tumor-specific MMP2/MMP9-mediated digestion, loaded Pep-4 neo-peptide was released, and competitively bound to HLA molecule on tumor cells. In the second step, Pep-4-specific TCR-T cells were adoptively transferred. By binding exogenous Pep-4 neo-peptide, HLA-A*2402-positive SYT-SSX fusion mutation-negative tumor cells were recognized by Tcr-T1 cells, eventually resulting in the eradication.

## Tumor-targeted delivery of neo-peptide by using stimuli-responsive nanoparticles

The stimuli-responsive nanoparticles were synthesized as described before (Liu et al, 2013). Briefly, peptide (PVGLIG, recognized by

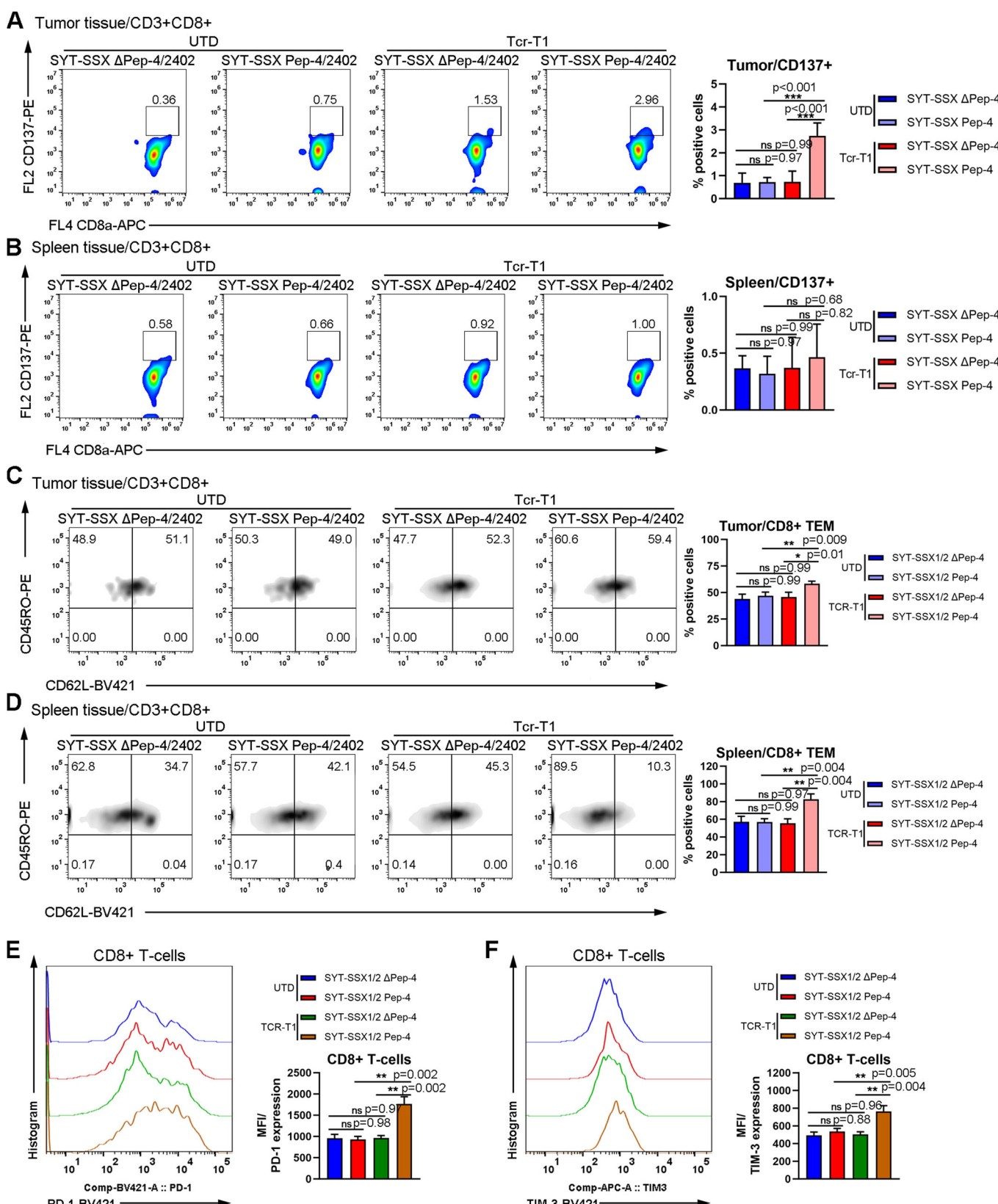

**Figure 4. Immunophenotypic analysis of transferred T cells.**

(A, B) Tumor tissues were isolated from mice of each group, and digested by using collagenases to prepare single-cell suspension. Flow cytometric analysis was performed to identify the immunophenotype of transferred T cells derived from tumor tissues of each group. Characterization of antigen-specific activation of T cells, defined by CD137 expression, derived from tumor tissue (A) and spleen (B). Representative flow cytometric image of CD137 expression on CD3 + CD8 + T cells in tumor tissues (A, left), and spleen (B, left) (n = 3 mice). The percentage of CD137-positive T cells in tumor tissues (A, right) and spleen (B, right) was summarized. (C, D) Flow cytometric analysis of effector memory T-cell (TEM) differentiation, characterized as CD45RO + CD62L- (n = 3 mice). Representative flow cytometric image of detecting TEM T cells in tumor tissues (C, left), and spleen (D, left) (n = 3 mice). The percentage of TEM in tumor tissues (C, right) and spleen (D, right) was summarized. (E, F) Representative flow cytometric image of T-cell exhaustion marker PD-1, and TIM-3 expression on CD3 + CD8 + T cells in tumor tissues (n = 3 mice). MFI was summarized. Data are presented as mean ± SEM. Statistical significance was calculated using the one-way ANOVA (A–F). ns not significant, *P < 0.05; **P < 0.01; ***P < 0.001. Source data are available online for this figure.

MMP2/9 (Zhu et al, 2022)) was PEGylated by using methoxy poly (ethylene glycol)-NHS (MePEG-NHS). Subsequently, mPEG-PVGLIG-PCL copolymer was synthesized via ring-opening copolymerization and double amidation (Li et al, 2009). Pep-4-loaded NPs (Pep4 NP) were prepared according to the nanoprecipitation method (Li et al, 2013). Pep-4 NP was identified as a hollow near spherical construction by using the transmission electron microscope, and the average hydrodynamic diameter was approximately 342 nm with a cationic surface charge (7.86±1.09 mV) detected by dynamic light scattering (DLS) (Fig. 5B,C). Importantly, the diameter and loading capacity of Pep-4 NP were almost unchanged compared with those freshly prepared (Fig. 5B; Appendix Table S3), indicating favorable stability. To monitor the tumor-targeted neo-peptide delivery of MMP2/MMP9-responsive Pep-4 NP, a Cy5.5-labeled Pep-4 was applied to generate NPs (Pep-4 Cy5.5 NP) as described before. As shown in Fig. 5D, when incubating Pep-4 Cy5.5 NP with MKN-45 highly expressing gelatinase (MMP2 and MMP9) (Sun et al, 2017), Cy5.5-labeled Pep-4 was rapidly released and bound to HLA-A*2402 molecule on MKN45 membrane surface. However, this process was dominantly abrogated by using an MMP2/MMP9 selective inhibitor, S 3304 (Chiappori et al, 2007). The in-vitro assay indicated that the generated Pep-4 Cy5.5 NP was able to deliver Pep-4 neo-peptide to HLA-A*2402 tumor cells in the presence of MMP2/MMP9. For further in vivo evaluation, we intravenously injected Pep-4 Cy5.5 NP into tumor-bearing mice formed by MKN-45 cells and monitored the tissue distribution via a living image system. We noticed that the fluorescent signal was detectable all over the body after injection, and significantly enriched in tumor tissue 24 h later (Fig. 5E), which persisted at least for 96 h (Fig. 5F). Interestingly, in the control group, mice were injected with NP loading Pep-2 identified unable to induce NRT cells (Fig. 1F), and 1 h later the fluorescent signal was similarly slightly higher in tumor tissue, which was undetectable in 48 h later (Fig. 5E). For further examination, the tumor tissue and other normal tissues, including heart, liver, lung, kidney and spleen, were isolated, and the content of Pep-4 in each tissue was analyzed by measuring total radiant efficiency. As expected, the fluorescent signal in tumor tissue from mice with Pep-4 Cy5.5 NP injection was much higher than that in other normal tissues (Fig. 5F). Possibly, Pep-4 Cy5.5 NP was passively accumulated in tumor tissue (Fig. 5G), binding with HLA-A*2402 molecule on tumor cells which protected it from elimination. The above results suggested that Pep-4 NP successfully mediated tumor-targeted in vivo delivery of neo-peptide to tumor cells without endogenous SYT-SSX fusion mutation, providing the feasibility to cooperate with Tcr-T1 cells.

## The cooperative therapeutic strategy mediated robust tumor regression in solid tumors

Next, we aimed to evaluate the therapeutic efficacy of the cooperative remedy combining Pep-4 NP with Tcr-T1 cells adoptive transfer. To this end, we firstly evaluated the in vivo anti-tumoral activity in mice bearing MKN-45 xenograft tumor model (Fig. 6A). When tumor volume reached nearly 100 mm³, tumor-bearing mice were injected with Pep-4 NP, and then were treated by adoptive transfer of Tcr-T1 cells one day later as illustrated in Fig. 6A. As expected, adoptive transfer of Tcr-T1 cells cooperated with Pep-4 NP elicited robust anti-tumoral activity in both tumor regression and prolonging survival (Fig. 6A–D), without apparent side effects (Fig. 6E). To furtherly investigate the specificity, we performed the mouse model, that one mouse bore two types of tumors side-by-side. As shown in Fig. 6F, consistent with our previous findings, adoptive transferring Tcr-T1 could specifically attenuate HLA-A*2402-positive MKN-45 tumor growth dependent on Pep-4 NP administration, while having little effect on HLA-A*0201-positive MDA-MB-231 cells.

To better simulate clinical practice, a disseminated peritoneal tumor model was established using MKN-45 transfected with luciferase activity. According to dynamic bioluminescence imaging, similar results were observed that the bioluminescence signal was significantly reduced only in mice treated with cooperative remedy (Fig. 6G). These data indicated that Pep-4 NP-mediated tumor-targeted delivery of neo-peptide successfully provided targets for Tcr-T1 cells, and functioned to trigger the recognition, engagement, and final tumor elimination independent of tumor inherent mutation profiles, with favorable safety, feasibility, and efficiency.

## Discussion

An ideal immunogenic tumor antigen is crucial to develop an effective cancer immunotherapy. Remarkably, gene fusions have been widely reported to show great potential to produce tumor neoantigens with higher immunogenic potential than other kinds of neoantigens (Wang et al, 2021). In this study, we identified a conservative, HLA-A*2402-restricted neo-peptide, Pep-4, shared by all SYT-SSX isoforms in almost all patients with synovial sarcomas. Using TCR repertoire sequencing and transcriptomic analysis, we verified a TCR, Tcr-1, that was of great avidity to enable primary T cells to specifically elicit Pep-4-specific HLA-A*2402-restricted cytotoxicity against SYT-SSX fusion mutation-positive synovial sarcoma cells both in vitro and in vivo.

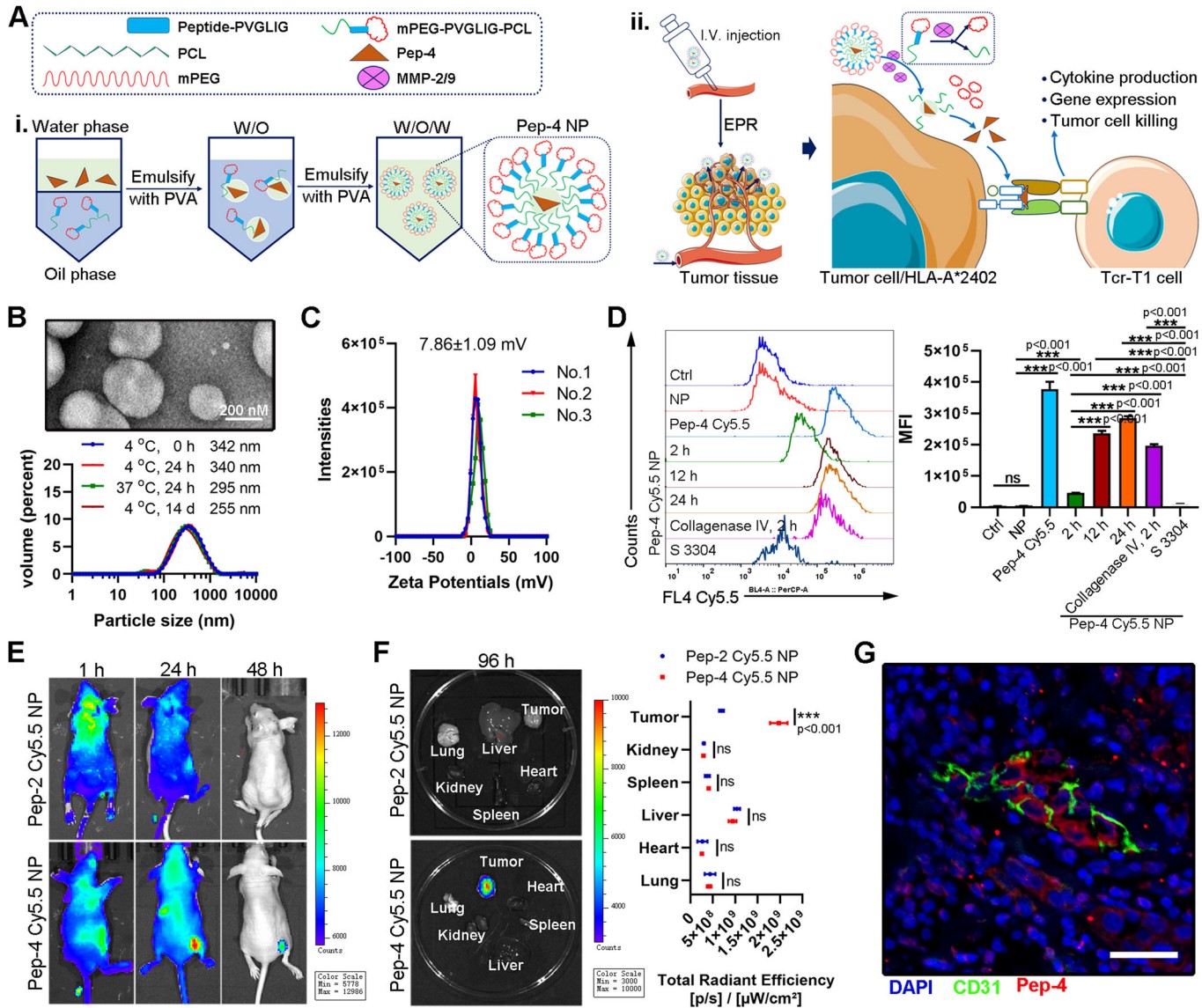

**Figure 5. Biodistribution and stimuli-responsive tumor targeting the delivery of Pep-4 NPs.**

(A) Schematic illustration of the preparation of Pep-4 NPs and action mode of cooperative therapeutic strategy via Pep-4 and Tcr-T1 cells. Abbreviation: mPEG, methoxy poly (ethylene glycol); PCL, polycaprolactone; PVA, polyvinyl alcohol. (B) Particle size distribution of Pep-4 NPs, freshly prepared, stored at 4 °C for 24 h, and for 14 d, and stored at 37 °C for 24 h. (C) Surface charge distribution of Pep-4 NPs. (D) Identification of stimuli-responsive tumor-targeted delivery of Pep-4. MKN45 cells ($5 \times 10^5$ cells/well) were incubated with NP (for 24 h), naked Pep-4 Cy5.5 (for 24 h), Pep-4 Cy5.5 NP (for 2, 12, 24 h), as well as collagenase IV-pretreated Pep-4 Cy5.5 NP (for 2 h). S 3304 was used to specifically suppress MMP-2/MMP-9. Representative flow cytometric images of analysis of Pep-4 Cy5.5 release and binding to MKN45 (left) and MFI were summarized ($n = 3$ independent replication) (right). (E, F) Subcutaneous tumor models were performed as described before. Pep-4 Cy5.5 NP was injected intravenously. NP containing Pep-2, identified as unable to induce NRT cells, was used as a control. Representative bioluminescence images of xenograft mice on 1 h, 24 h, and 48 h after NP injection (E). 96 h later, tumor-bearing mice were sacrificed and liver, lung, heart, kidney, spleen, and tumor tissue were isolated. Bioluminescence imaging detected NP enrichment in different tissues (F, left). Quantification of accumulated NPs from the indicated tissues ($n = 3$ mice) (F, right). (G) The representative fluorescent image revealed the localization of released Pep-4 Cy5.5 (red) and tumor vessel (green). Nuclei were stained with DAPI. Scale bar = 20 μm. Data are presented as mean ± SEM. Statistical significance was calculated using the one-way ANOVA (D), and two-way ANOVA (G). ns not significant, *$P < 0.05$; **$P < 0.01$; ***$P < 0.001$. Source data are available online for this figure.

Furthermore, cooperating with a stimuli-responsive tumor-targeted delivery system extended the application of Tcr-T1 cell therapy as a functional TCR-T therapeutic regimen for HLA-A*2402-positive solid tumors.

ACT has changed the landscape of cancer immunotherapy (Baulu et al, 2023). Adoptive transfer of T cells, including TCR-T

cell therapy as well as tumor-infiltrating lymphocyte (TIL) therapy has shown great promise, generating unprecedented response rates to treatments of patients with malignancies (Fesnak et al, 2016). However, high heterogeneity, scarcity of available antigens, and immunosuppressive tumor microenvironment provide dominant limitation of CAR-T clinical efficacy in solid tumors (Fesnak et al,

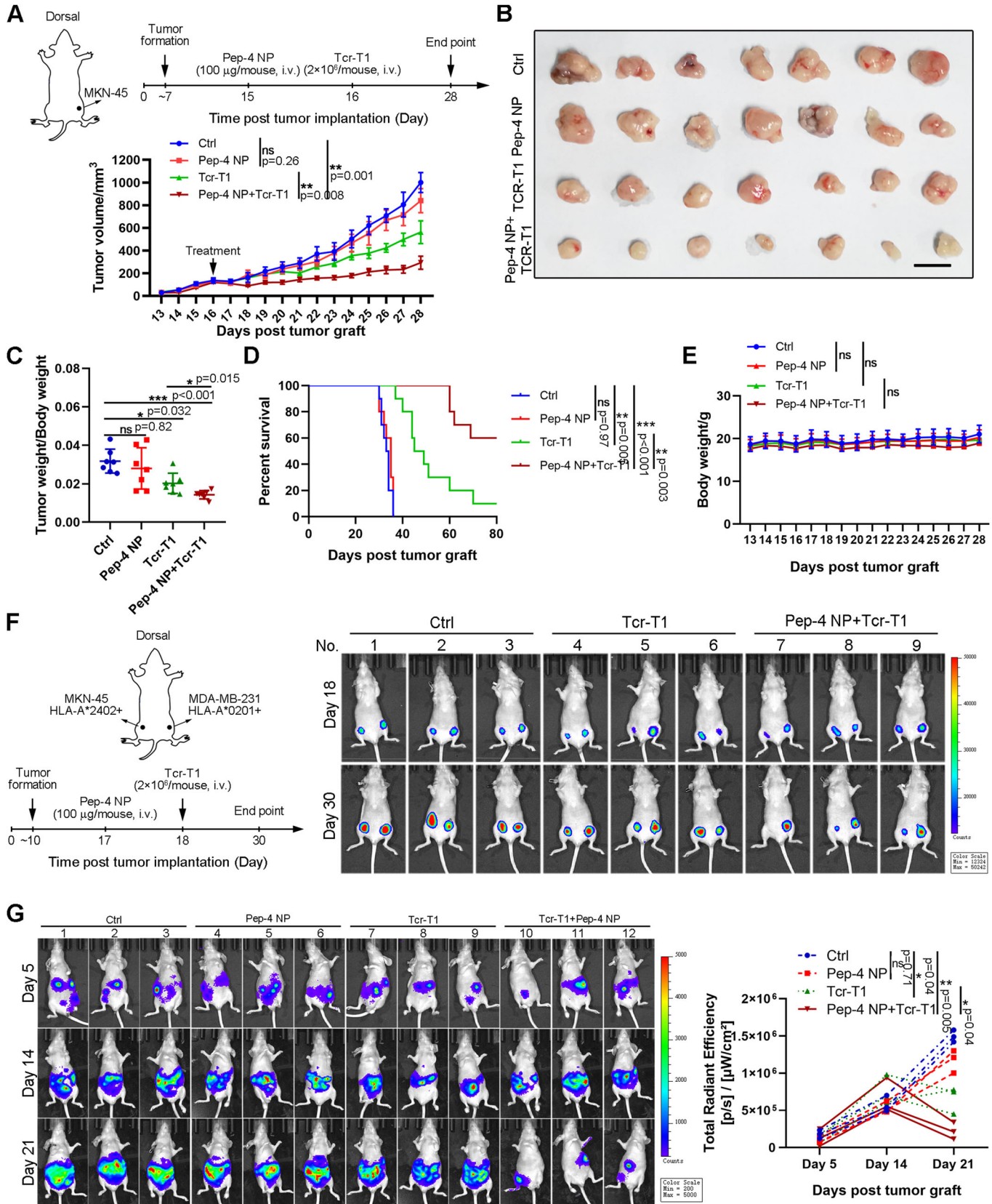

**Figure 6.  The cooperative therapeutic strategy exhibited robust anti-tumoral activity against HLA-A*2402-positive solid tumors.**

(A) Treatment scheme (top). HLA-A*2402-positive MKN45 cells were used to perform subcutaneous tumor models. When tumor volume reached nearly 100 mm³, tumor-bearing mice were injected with Pep-4 NP and then were treated by adoptive transfer of Tcr-T1 cells one day later (n = 7 mice). Tumor volumes of each group were recorded (bottom). (B–E) Mice from (A) were sacrificed at the endpoint. Representative photographs of tumor tissues isolated from each group of mice (B), Tumor weights were recorded and the ratio of tumor weight versus body weight was presented (n = 7 mice) (C). The overall survival of each group was analyzed using the Kaplan–Meier survival analysis (D). The body weights of mice from each group were recorded (n = 7 mice) (E). (F) The mouse model that one mouse bearing two types of tumors side-by-side was performed. HLA-A*2402-positive MKN-45 cells and HLA-A*0201-positive MDA-MB-231 cells were subcutaneously implanted in the opposite of one mouse. The cooperative therapeutic regimen was performed as indicated. The tumor volume was summarized via living imaging. Representative bioluminescence images of xenograft mice on Day 18, and Day 30. (G) A disseminated peritoneal MNK45 tumor-bearing mouse model was performed as mentioned in methods (n = 3 independent replication). 5 days after tumor injection, a cooperative therapeutic regimen was performed as described in (A). Representative bioluminescence images of xenograft mice on Day 5, Day 14, and Day 21 (left). Tumor growth profiles were quantified (right). Data are presented as mean ± SEM. Statistical significance was calculated using the one-way ANOVA (C), and two-way ANOVA (A, E, F). Survival curves were analyzed with a log-rank test (D). ns not significant, *$P < 0.05$; **$P < 0.01$; ***$P < 0.001$. Source data are available online for this figure.

2016). Alternatively, TCR-T cells are capable of recognizing epitopes in a much lower density required for T cell activation than classical CAR-T cells (Fesnak et al, 2016). To date, many clinical trials have illustrated an impressive therapeutic effect of TCR-T cell therapy against various solid tumors by targeting MAGEA3/A4 (Lu et al, 2015), NY-ESO-1 (Robbins et al, 2015; Robbins et al, 2011a), MART1 (Chodon et al, 2014), and viral antigen HPV16-E6 (Doran et al, 2019). Unfortunately, MAGE-A3 TCR-T cells were observed not only to attack MAGE-A3-positive tumor cells but also exhibit cytotoxicity against the normal central nervous system through the cross-reactivity to MAGE-A12, which is widely expressed in the human brain (Cameron et al, 2013; Linette et al, 2013). Thus, figuring out tumor-specific antigens (TSAs), those specifically expressed in tumor cells and efficient in inducing antigen-specific activation of T cells, is vital for developing TCR-T cell therapy. Neoantigens produced from peptides derived from cancer-specific aberrant proteins undergoing site mutation, alternative splicing, and particularly genomic fusion are regarded as important sources of TSAs (Wang et al, 2021). Given that gene fusion in tumor cells dominantly occurs in the key transcription factors (Zhou et al, 2022) responsible for 20% of global cancer morbidities, neoantigens derived from genomic fusion can be ideal targets for cancer immunotherapy. Indeed, in this study, we identified a conserved fusion mutation neo-peptide, Pep-4, shared by all SYT-SSX isoforms, which was sufficient to induce neoantigen-reactive T-cell consistent with previous findings (Kawaguchi et al, 2005). These findings highlight that the shared fusion mutation neo-peptide Pep-4 is of great potential as an ideal target.

The identification of neoantigens with high immunogenicity is very important to develop highly personalized TCR-T therapy. Lucca and colleagues revealed that circulating T cells shared similar clonal expansion, and gene signatures of effector functions, but not terminal exhaustion with NRT cells in tumor tissues, providing a piece of solid evidence to screen TSA-specific TCR from peripheral T cells (Lucca et al, 2021). Great development in silico prediction algorithms and deep-sequencing technologies make it possible to rapidly predict candidate neoepitopes. However, tremendous challenges remain to be solved. In this study, we and others (Kawaguchi et al, 2005) observed that Pep-4 could bind with HLA-A*2402 and induce a neoantigen-specific activation of NRTs characterized by ELISpot assay (Fig. 1F). However, Pep-4 was predicted with no binding activity against any one of indicated HLA-A molecules according to NetMHC-pan4.0, indicating a

particular deviation between the online prediction and the actual situation.

Apart from the difficulty in potential neo-peptide identification, a practical procedure for rapidly raising large numbers of neoantigen-specific T-cells is historically another technical bottleneck during the clinical application of TCR-T cell therapy. Traditionally, monocytes are firstly isolated from PBMC of healthy donors and differentiated into professional antigen-presenting dendritic cells with exposure to GM-CSF. Next, naive T cells are incubated with matured DC cells with neoantigen pulsing, and subsequently expanded weekly by using cytokines for further TCR sequencing and functional analysis (Ali et al, 2019). However, due to the extra steps for isolation and clonal expansion, it is time-consuming and labor-consuming. In this study, we optimized the two-step culture for fast expansion of NRT cells by using unfractionated PBMCs as described before (Pathangey et al, 2017), and successfully induced within 2 weeks (Fig. 1C).

Recently, multiple lines of evidence have discovered that NRT cells are enriched in some T-cell subclones characterized by high expressions of CD103 (Duhen et al, 2018), CD137 (Eiva et al, 2022), PD-1 (Purcarea et al, 2022) or CD39 (Chatani et al, 2023). Additionally, by the comprehensive application of single-cell RNA sequencing and TCR sequencing analysis, Caushi and colleagues noticed that tissue-resident memory T-cell signature genes, ZNF683 and ITGAE, were shared in most NRT subclones, although they presented a significant diversification of phenotype (Caushi et al, 2021). More importantly, they also verified that CXCL13 was a functional marker for both CD8+ and CD4 + NRT cells (He et al, 2022). Consistent with previous findings (Meng et al, 2023), we observed that Tcr-1-positive CD8+ subclone was identified as high expression of GNLY, ZNF683, CXCL13 (Fig. 2H, and Appendix Fig. S3G), and were furtherly proven of high avidity against SYT-SSX fusion-positive tumor cells (Appendix Fig. S3G). Thus, better identification of gene expression profiling and corresponding TCR repertoire in each T-cell using sing-cell sequencing technology might provide a more precise prediction for high avidity TCR selection.

Due to the high heterogeneity of solid tumors and HLA restriction, TCR-T cell therapy is highly personalized resulting in remarkable cost and manufacturing complexity during clinical application (Shafer et al, 2022). Interestingly, intratumoral injection of neo-peptides can effectively induce neo-peptide-HLA complex formation, and direct NRT cells to show cytotoxicity against tumor cells without neoantigen expression (Nobuoka et al, 2013).

Similarly, in our previous work, the antigen-loaded fusogenic nanoparticles (F-AgNPs) successfully delivered antigens to tumor tissue, and retargeted CAR-T cells against tumor cells in response to delivered antigens (Sun et al, 2022). These studies pave the way for us to develop a TCR-T cell therapy-based cooperative therapeutic strategy for solid tumors. Indeed, by using the well-defined stimuli-responsive tumor-targeted delivery system (Li et al, 2013), Pep-4 was successfully delivered into tumor tissue, and sufficiently triggered Tcr-T1 cells against MKN-45 cells without SYT-SSX fusion mutation, resulting in a remarkable suppression on tumor growth in several tumor models, while no obvious side-effects were observed.

Although the cooperative TCR-T therapy showed impressive efficacy against HLA-A*2402-positive tumor cells in various tumor-bearing mouse models, we could not deny that a notable limitation existed in this study. Due to the lack of commercial humanized mice, those that express human HLA-A*2402 molecule and are immune deficient parallelly, the potential on-target off-tumor side effects could not be completely investigated. Thus, a clinical investigation, such as an investigator-initiated trial, is particularly urgent. However, it is a time-consuming and costly process.

Collectively, in this study, we not only discovered paired TCRs that specifically targeted gene fusion neoantigen but also developed a therapeutic strategy for solid tumors independent of inherent mutation profiling, providing a new perspective for the future development of TCR-T cell therapy for clinical application.

# Methods

### Reagents and tools table

| Reagent/resource | Reference or source | Identifier or catalog number |
|---|---|---|
| **Experimental models** | | |
| SW982 cell (H. sapiens) | typical cell culture collection of the Committee of the Chinese Academy of Sciences Library (Shanghai, China) | TCHu209 |
| MDA-MB-231 (H. sapiens) | typical cell culture collection of the Committee of the Chinese Academy of Sciences Library (Shanghai, China) | TCHu227 |
| MKN45 (H. sapiens) | China Center for Type Culture Collection | GDC0220 |
| T2 (H. sapiens) | ATCC | CRL-1992 |
| T2-A24 cells (H. sapiens) | This study | N/A |
| SW982-pMHCΔPep4 (H. sapiens) | This study | N/A |
| SW982-pMHC (H. sapiens) | This study | N/A |
| **Recombinant DNA** | | |
| piggyBAC-U6-GFP-WPRE | Addgene | #200030 |
| PB-CMV-MCS-Puro | Addgene | #219794 |

| Reagent/resource | Reference or source | Identifier or catalog number |
|---|---|---|
| pDonor-SB-Tcr-1-RQR8-EGFP | This study | N/A |
| pDonor-SB-Tcr-2-RQR8-EGFP | This study | N/A |
| pDonor-SB-Tcr-3-RQR8-EGFP | This study | N/A |
| pDonor-SB-Tcr-4-RQR8-EGFP | This study | N/A |
| pDonor-SB-SYT-SSX-EGFP | This study | N/A |
| pDonor-SB-SYT-SSXΔPep4-RQR8-EGFP | This study | N/A |
| pDonor-SB-SYT-SSX/2402-EGFP | This study | N/A |
| pDonor-SB-SYT-SSXΔPep4/2402-EGFP | This study | N/A |
| **Antibodies** | | |
| CD3-FITC | Miltenyi Biotec | Cat no. 170-081-047 |
| CD4-PerCP | Miltenyi Biotec | Cat no. 130-113-779 |
| CD8-APC | Miltenyi Biotec | Cat no. 130-113-154 |
| CD137-PE | Miltenyi Biotec | Cat no. 130-119-885 |
| CD3-BV510 | BD Biosciences | Cat no. 564713 |
| CD4-APC-Cy7 | BD Biosciences | Cat no. 561839 |
| CD8-BV605 | BD Biosciences | Cat no. 564116 |
| CD45RO-PE | BD Biosciences | Cat no. 561889 |
| CD62L-BV421 | BD Biosciences | Cat no. 563862 |
| TIM-3-Alexa 647 | BD Biosciences | Cat no. 565559 |
| **Chemicals, enzymes, and other reagents** | | |
| Human GM-CSF | Genscript | Cat no. Z02190-50 |
| Human IFN-α 2 A | Genscript | Cat no. Z03003-50 |
| Human IL-7 | Genscript | Cat no. Z02704-10 |
| Resiguimod (R848) | MCE | Cat no. HY-13740 |
| LPS | MCE | Cat no. HY-D1056 |
| RPMI-1640 | Invitrogen | 11875093 |
| DMEM | Invitrogen | 11965092 |
| Fetal bovine serum (FBS) | Gibco | 10099 |

| Reagent/resource | Reference or source | Identifier or catalog number |
| --- | --- | --- |
| AIM-V medium | Gibco | Cat no. 12055083 |
| penicillin-streptomycin | Beyotime | C0222 |
| BCA protein assay | Beyotime | P0010 |
| Amaxa Human T cells Nucleofector Kit | Lonza | VPA-1002 |
| Single Cell 5' Library and Gel Bead Kit | 10X Genomics | Cat no. 1000006 |
| Chromium Single A Chip Kit | 10X Genomics | Cat no. 120236 |
| **Software** | | |
| GraphPad Prism 10 | https://www.graphpad.com/ | |
| FlowJo | https://www.flowjo.com/ | |
| **Other** | | |
| BD FACS LSRFortessa™ | BD | |
| Zeiss LSM 880 with_Airyscan | Zeiss | |

## Methods and protocols

### Cell culture

Human synovial sarcoma cell line SW982 (HLA-A24-, SYT/SSX fusion-negative), human gastric adenocarcinoma cell line MKN45 (HLA-A24 +, SYT/SSX fusion-negative), human breast cancer cell line MDA-MB-231 (HLA-A02 +, SYT/SSX fusion-negative) and human transporter-associated protein (TAP)-deficient cell line T2 was originally purchased from ATCC, China Center for Type Culture Collection (CCTCC) and the typical cell culture collection of the Committee of the Chinese Academy of Sciences Library. All cell lines were cultured in RPMI-1640 medium or DMEM medium (Invitrogen) supplemented with 10% FBS (Gibco, Cat no. 10099) and 1% penicillin-streptomycin at 37 °C with 5% $CO_2$ according to the supplier's instructions. SW982-pMHC and SW982-pMHCΔPep4 cell lines, representing the transfected SW982 cell lines stably expressing peptide-MHC multimer (pMHC) or MHC monomer respectively, were generated in our lab. For peptide affinity analysis, the T2 cell line was transduced with HLA-A*2402 cDNA (T2-A24 cells). All cells tested negative for mycoplasma and were authenticated by short tandem repeat DNA fingerprinting at the Key Laboratory of Pharmaceutical Biotechnology and The Comprehensive Cancer Center of Nanjing University (Nanjing, China). No cell lines were passaged for more than 6 months after resuscitation in our study.

### Fluorescence in situ hybridization (FISH) assay

To detect SYT-SSX fusion mutation, a FISH assay was performed as described before (Tay et al, 2021). Briefly, FFPE sections were baked at 56 °C overnight followed by deparaffinization in xylene. After dehydration through an ethanol series, the slides were treated with sodium thiocyanate and protease digestion. Dual-color break-apart rearrangement probe was applied to tumor tissue, followed by overnight hybridization at 37 °C. The first probe was labeled in SpectrumOrange™ (650 kb, telomeric, 5' to SYT), and the second probe was labeled in SpectrumGreen™ (1044 kb, centromeric, 3' to SSX). The slides were washed and counterstained with 4', 6-diamidino-2-phenylindole (DAPI) anti-fade solution and analyzed under an epifluorescence microscope.

### SYT/SSX fusion neoantigen peptides and corresponding HLA binding affinity prediction

Essentially in all synovial sarcomas, a t(X;18) represents the fusion of SYT (at 18q11) with either SSX1 or SSX2 (both at Xp11) (Ladanyi, 2001). Whole genomic sequencing (WGS) and fluorescence in situ hybridization (FISH) revealed that SYT-SSX1 and SYT-SSX2 shared a similar amino acid sequence near the breakpoint. The shared 21-mer peptide (PQPPQQRPYGYDQIMPKKPAE) was subjected to NetMHC-pan4.0 epitope-HLA prediction algorithm (Andreatta and Nielsen, 2016) for SYT/SSX fusion neo-peptides and corresponding HLA (top ten in Chinese (He et al, 2018)) binding affinity prediction. The predicted neo-peptides were synthesized (GenScript, purification >95%), and stored at −80 °C until needed.

### Optimized two-step culture protocol for fast in-vitro expansion of NRT cells

Firstly, HLA-A*0201 +, HLA-A*1101 +, and HLA-A*2402+ eligible volunteer donors ($n = 3$ for each group) consented to undergo repetitive leukapheresis collections at intervals deemed safe. The blood collection procedure was carried out following the guidelines verified and approved by the Ethics Committee of Nanjing Drum Tower Hospital. All donors signed an informed consent for the scientific research statement. Peripheral blood mononuclear cells (PBMCs) were isolated from samples from healthy donors by centrifugation on a Ficoll density gradient and suspended in an AIM-V medium (Gibco, Cat no. 12055083). And then, the unfractionated PBMC was then subjected to a two-step culture for fast expansion of antigen-specific T cells as described before (Pathangey et al, 2016). Briefly, PBMCs ($10^6$ cells/ml) were seeded in a 24-well plate. As shown in Fig. 1C, agents added standardly during Step 1 (Day 0-2 of culture) included recombinant human GM-CSF (rhGM-CSF, 1000 IU/ml, on Day 0; Genscript, Cat no. Z02190-50), recombinant human IFN-α 2 A (10 ng/ml, on Day 0; Genscript, Cat no. Z03003-50), Resiquimod (R848, 3 μg/ml, on Day 1; MCE, Cat no. HY-13740), LPS (50 ng/ml, on Day 1; MCE, Cat no. HY-D1056). Predicted neo-peptide Pep-1 (YGYDQIMPK), Pep-2 (YGYDQIMPKK), Pep-3 (RPYGYDQIMPK), and Pep-4 (GYDQIMPKK) were synthesized by GenScript Company (Nanjing, China), and were added on Day 3 (10 μg/ml). Next, recombinant human IL-7 (10 ng/ml; Genscript, Cat no. Z02704-10) was added during Step 2 (Day 5 to ~14).

### In-vitro expansion of antigen-specific T cells

On day ~14 of stimulation, fusion neoantigen-specific T cell clones were characterized by evaluating IFN-γ secretion via CBA. Specific T-cell clones were further cultured in AIM-V medium with 10%

human AB serum (Valley biomedical, Cat no. HP1022), 50 ng/ml OKT-3 (eBioscience, Cat no. 14-0037-82), 20 IU/ml rhIL-2 (Genscript, Cat no. Z00368-50), 12.5 ng/ml rhIL-7, and 12.5 ng/ml rhIL-15 (Genscript, Cat no. Z00377-50).

## Single-cell transcriptomic analysis and TCR repertoire sequencing

### Preparation of single-cell suspensions

Neoantigen-responsive T-cell clones were harvested and CD8 + T cells were furtherly isolated through magnetic beads (Miltenyi Biotec, CD8 + T cell Isolation Kit, Cat no. 130-096-495). Purified CD8 + T cells were subjected to single-cell RNA sequencing (scRNA-seq) for transcriptomic analysis and TCR sequencing as reported (Zhang et al, 2020).

### Droplet-based single-cell sequencing

Briefly, using a Single Cell 5' Library and Gel Bead Kit (10X Genomics, Cat no. 1000006) and Chromium Single A Chip Kit (10X Genomics, Cat no. 120236), the single cell suspension (300–600 living cells/ml) was loaded onto a Chromium single cell controller (10X Genomics) to generate single-cell gel beads in the emulsion (GEMs) according to the manufacturer's protocol. Single cells were suspended in PBS containing 0.04% BSA. Approximately 10,000 cells were added to each channel and approximately 6000 target cells were recovered. Captured cells were lysed and the released RNA was barcoded through reverse transcription in individual GEMs. Reverse transcription was performed on an S1000TM Touch Thermal Cycler (Bio-Rad) at 53 °C for 45 min, followed by 85 °C for 5 min and a hold at 4 °C. Complementary DNA was generated and amplified, after which, quality was assessed using an Agilent 4200 (performed by CapitalBio Technology). According to the manufacturer's introduction, scRNA-seq libraries were constructed using a Single Cell 5' Library and Gel Bead kit, Human T Cell (Cat no. 1000005). The libraries were sequenced using an Illumina Novaseq6000 sequencer with a paired-end 150-bp (PE150) reading strategy (performed by CapitalBio Technology). Raw data were deposited in the public database GEO (accession no. GSE243535).

### Sc-TCR sequencing

Full-length TCR V(D)J segments were enriched from amplified cDNA from 5' libraries via PCR amplification using a Chromium Single-Cell V(D)J Enrichment kit according to the manufacturer's protocol (10X Genomics). For TCR, cells with at least one productive TCR α-chain (TRA) and one productive TCR β-chain (TRB) were kept for further analysis.

## Plasmids construction

The piggyBac transposase plasmid (piggyBAC-U6-GFP-WPRE) and corresponding transposon backbone plasmid (PB-CMV-MCS-Puro) were purchased from Addgene. As shown in Appendix Fig. S4A, the human TRA and TRB sequences were linked by the 2A sequence, and the constant regions were replaced by mouse constant regions to increase the TCR transduction level. Additionally, an RQR8 (Philip et al, 2014), a highly compact marker/suicide gene containing CD34 and CD20 epitopes, was co-expressed with TCR by cloning it downstream of the TCR separated by the 2A

sequence. EGFP was also cloned downstream of RQR8. Annotated sequences of constructs used are included in Appendix Fig. S4A and Appendix Table S2.

## Generation of TCR-T cells

PBMCs were isolated from healthy volunteers, and suspended in AIM-V medium for two hours, and non-adherent cells were harvested and expanded in a complete medium containing 90% AIM-V medium, 10% FBS serum, subsequently activated with 50 ng/ml OKT-3, and 100 IU/ml rhIL-2. After 48 h stimulation, OKT-3 activated T cells ($10^7$ cells/kit) were transfected with the intended plasmids (5 μg/kit piggyBac transposase plasmid and 5 μg/kit TCR transposon plasmid) by using Nucleofector 2B (Lonza, Germany). Briefly, $10^7$ cells were washed with PBS twice and resuspended in 100 μl transfection buffer (Amaxa Human T cells Nucleofector Kit, VPA-1002, Lonza, Germany), mixed with plasmids. Program T-007 was selected for electroporation. After transfection, cells were resuspended in pre-warmed AIM-V medium, and the cell culture medium was half replaced by fresh complete medium containing 20 IU/ml rhIL-2, 12.5 ng/ml rhIL-7, and 12.5 ng/ml rhIL-15 every 2–3 days.

## TCR-T specificity

Stably HLA-A*2402-expressed transgenic T2 cells (T2-A24), SW982-pMHCΔPep4, SW982-pMHC, and HLA-A*2402+/SYT-SSX- MKN45 cells were used as target cells. Transgenic TCR-T cells were co-incubated with fusion neoantigen-loaded (10 μM) target cells ($5 \times 10^4$ cells/well) in effector-to-target ratios (E: T) ranging from 0.2:1 to 5:1. After 24 h of co-culture supernatants were harvested and analyzed by IFN-γ CBA, and cell viability was evaluated by luciferase-based cell viability examination.

## Cell viability determination

Target cells (SW982, and MKN45) were labeled via stable expression of luciferase (SW982-luci, and MKN45-luci), which were applied for further determination of TCR-T cytotoxicity. As described before, target cells were incubated with transgenic TCR-T cells in effector-to-target ratios ranging from 0.2:1 to 5:1. After 24 h of co-culture, target cell viability was determined by using optimized luciferase assay according to previous findings (Matta et al, 2018). Briefly, detection buffer containing 1 mM D-Luciferin (Yeasen, cat no. 40901ES03), 25 mM Gly-Gly, 15 mM potassium phosphate, 15 mM MgSO$_4$, 4 mM EGTA, 2 mM ATP, and 1 mM DTT, were added to cell culture medium at ratio of 1:10 (v/v). Luminescence was read in endpoint mode using the Tecan Infinite F Plex reader. Percentage Cell viability was calculated using the luciferase activity of non-treated cells as the maximum cell survival and blank well as background control, and using the formula % cell viability = 100 × [(experimental data − background control)/(maximum cell survival − background control)].

## HLA-A24 peptide affinity assay

HLA-A24 specific binding for each peptide was evaluated by flow cytometry using the T2-A24 cell line. In the assay, T2-A24 cells were washed in serum-free AIM-V medium and resuspended to a

final concentration of $1 \times 10^6$ cells/ml. The cells were pulsed with various concentrations (0–200 µg/ml) of respective synthetic SYX/SSX fusion peptides (50 µg/ml) plus human β2-microglobulin (3 µg/ml) (Merck, Cat no. SAE0112), and then incubated at 26–28 °C in 5% $CO_2$ humidified air. Following overnight incubation, the cells were washed in PBS containing 5% FBS, stained with mouse anti-human HLA-A24-PE mAb (LSBio, clone 22E1, cat no. LS-C179737), and analyzed by using flow cytometry.

## IFN-γ ELISpot assay procedure

Human IFN-γ ELISpot assay was established by using the Human IFN-γ ELISPOT Set (BD, Cat no. 551849) according to the manufacturer's instructions. Briefly, the pre-wetting ELISPOT plate was incubated with diluted coating antibody, and stored at 4 °C overnight. Using sterile conditions and an aseptic technique, the 96-well pre-coated assay plate was blocked with Blocking Solution (2 h, at RT). To each well of a blocked pre-coated ELISpot plate, 100 µl of stimulated T cells ($2.0 \times 10^6$ cells/ml) was added for a final cell density of $2.0 \times 10^5$ cells/well. T-cells were incubated for 24-48 h at 37 °C, 5% $CO_2$. Then, cells were removed from the plate and IFN-γ was detected using sequential incubations of a biotinylated anti-IFN-γ antibody and a streptavidin alkaline phosphatase-coupled detection antibody. Spots were visualized following the addition of 100 µl 5-bromo-4-chloro-3-indolyl phosphate/nitro blue tetrazolium (BCIP/NBT)-plus substrate for 3 min ± 15 s unless otherwise specified. Spot development was stopped using a water wash and plates were air-dried overnight at RT, avoiding exposure to light. SFU in each well was enumerated using an automated spot counter (ImmunoSpot CTL S6 Micro Analyzer, Cellular Technology Limited) within 24–96 h of development. The settings used on the immunespot to detect the secreted IFN-γ spots were sensitivity of 145, background balance of 10, spot separation of 1, counting mask size of 90% (not normalized), minimum spot size of 0.0015 mm², and a maximum spot size of 9.6466 mm². The spot counts in each well were visually quality controlled by an analyst to ensure that the spot detection was appropriate, before finalization.

## Flow cytometry

For the peptide affinity assay, single-cell suspensions of T2-A24 cells from each group were obtained with EDTA, blocked for 30 min on ice in PBS (with 0.2% IgG-free bovine serum albumin and 2 mM EDTA), and subsequently incubated with HLA-A24-PE mAb for 30 min on ice. The cells were washed and detected by using BD FACSCalibur 4 CLR. HLA-A24 peptide binding affinity was determined by measuring the level of up-regulation of HLA-A24 molecules on the T2-A24 cells induced by specific and stable peptide binding. HLA-A24 peptide binding affinity is reported as the median fluorescence intensity (MFI) of HLA-A24 expression. The results are shown as the mean MFI ± SEM.

For measurement of secreted IFN-γ by cytometric bead array (CBA), the supernatants of T-cells after indicated treatment were harvested, incubated with pre-coated beads according to the manufacturer's instructions (BD, Cat no. 558269), and then subjected to cytometric analysis. The results were expressed as mean fluorescence intensity, and the concentration was calculated according to the standard curve.

For phenotypic characterization of T cells, single-cell suspensions were prepared from tumor tissues. All cells were pre-incubated with FcR blocking antibody (FcR Blocking Reagent, Miltenyi Biotec, Cat no. 130-092-575) for 15 min at 4 °C at a concentration of 1 µg/ $1 \times 10^6$ cells/100 µl. Fluorescent antibodies (Miltenyi Biotec, CD3-FITC, Cat no. 170-081-047; CD4-PerCP, Cat no. 130-113-779; CD8-APC, Cat no. 130-113-154; CD137-PE, Cat no. 130-119-885. And BD Biosciences, CD3-BV510, Cat no. 564713; CD4-APC-Cy7, Cat no. 561839; CD8-BV605, Cat no. 564116; CD45RO-PE, Cat no. 561889; CD62L-BV421, Cat no. 563862; TIM-3-Alexa 647, Cat no. 565559) and isotype antibodies were consequently added at the indicated concentration, and cells were incubated for a further 30 min at 4 °C. Then cells were washed twice with ice-cold PBS. Flow cytometry analysis was performed with the BD FACSCalibur.

## Animal studies

The objective of in vivo efficacy studies was to evaluate the activity of Tcr-T1 cells in several tumor-bearing mouse models. The sample size ($n = 6$–7 mice per group) was determined based on the consistency and homogeneity of tumor growth in the various models and was sufficient to determine statistically significant differences in tumor response between the various treatment groups. Animals were randomized based on tumor size and were treated with various treatments until the average tumor volume reached 100 mm³. All animals were purchased from the Model Animal Research Center of Nanjing University (Nanjing, China). The animal studies were approved by the Laboratory Animal Welfare and Ethics Committee of Nanjing Drum Tower Hospital (IACUC-2109010) and carried out at Nanjing Drum Tower Hospital. All animals were housed under SPF conditions. All animal experiments conformed to the guidelines of the Animal Care and Use Committee of Nanjing Drum Tower Hospital. The investigation was not blinded for animal studies and all efforts to minimize suffering were made.

For subcutaneous tumor models, SW982-pMHCΔPep4, SW982-pMHC and HLA-A*2402⁺/SYT-SSX⁻ MKN45 cells ($5 \times 10^6$ cells) were resuspended in 200 ml PBS with 30% Matrigel (cat no. 356234, BD, America) and injected subcutaneously into the flanks of 6-week-old female BALB/c nude mice. In SW982-formed tumor-bearing mice, engineered Tcr-T1 cells ($5 \times 10^6$ cells/mouse) and UTD T-cells ($5 \times 10^6$ cells/mouse) were intravenously injected respectively, when the average tumor volumes reached 100 mm³. In HLA-A*2402⁺/SYT-SSX⁻ MKN45-formed tumor-bearing mice, neoantigen-loaded nanoparticles, Pep-4 NP (100 µg/mouse), or PBS were intravenously injected when the average tumor volumes reached 100 mm³. 24 h later, engineered Tcr-T1 cells ($5 \times 10^6$ cells/mouse) were intravenously transferred. Tumor volumes were measured using an electronic caliper and calculated using the following formula: Volume (mm³) = $L \times W^2 \times \pi/6$, where L and W represent the largest and smallest diameters, respectively. The animals were sacrificed in a $CO_2$ chamber in accordance with IACUC guidelines. Tumors were harvested and used for histological analysis or rapidly frozen. Body weights were recorded every day.

For the peritoneal metastasis tumor model, 6-week-old female BALB/c nude mice were injected intraperitoneally with MKN45 cells labeled with luciferase (MKN45-Luci). Nearly 5–7 days after

**The paper explained**

**Problem**

Adoptive T-cell transfer, in particular T-cell receptor-engineered T-cell (TCR-T) therapy, holds great promise for cancer immunotherapy with encouraging clinical results. TCR-T cell therapy reactivity is restricted to tumor antigens presented by human leukocyte antigen (HLA) molecules. Although tumor-associated antigens (TAAs) are widely used as the target for the development of TCR-T cell therapy in various malignancies, sufficient safety and effectiveness remain to be well documented due to the deficiency of their specificity. The engineered T-cell that targets neoantigens, especially fusion neo-peptides, is seldom reported.

**Results**

In this study, we first identified a functional TCR, Tcr-1, which selectively recognized the SYT-SSX fusion neoantigen shared by most synovial sarcomas. Engineered T-cell expressing Tcr-1 (Tcr-T1) demonstrated HLA-A*2402-restricted, antigen-specific anti-tumoral efficacy against synovial sarcoma cells, both in vitro and in vivo. Furthermore, we developed a cooperative therapeutic modality, in which exogenous SYT-SSX fusion neoantigen was loaded into stimuli-responsive nanoparticles (NPs) formed by mPEG-PVGLIG-PCL copolymers (Neo-AgNPs) for tumor targeting delivery. In situ, this modification was able to direct engineered Tcr-T1 to other HLA-A*2402-positive malignant cancer cell lines with significant antigen-specific cytotoxicity despite their inherent mutation profiles, which further extend its application.

**Impact**

We not only identified a functional TCR, which selectively recognized fusion neoantigen but also provided an inherent mutation-independent tumor-targeted cooperative TCR-T therapeutic regimen. This cooperative therapy provides a possible strategy to break through the limitations of TCR-T therapy, such as HLA-restriction, and mutational heterogeneity.

tumor implantation, the tumors could be detected via bioluminescence imaging. Then, the tumor-bearing mice were randomized into four groups. Pep-4 NP (100 μg/mouse), or PBS was intraperitoneally injected, when the average tumor volumes reached 100 mm³. 24 h later, engineered Tcr-T1 cells ($5 \times 10^6$ cells/mouse) were intraperitoneally transferred. Tumor burden was monitored on day 5, day 14, and day 21 after implantation, using the IVIS Lumina III system (PerkinElmer, Massachusetts, USA).

Animal care and experimental procedures were approved by the Institutional Animal Care and Use Committee, Nanjing University (Accreditation No. IACUC-2022023).

### Statistical analysis

All statistical analyses were performed by GraphPad Prism 10 (GraphPad). Differences between the two groups were calculated by Student's *t* test. Multiple comparisons between two populations were conducted by multiple *t* tests with type 1 error correction. Differences among multiple groups were calculated by one- or two-way ANOVA. Differences in survival were calculated by the log-rank Mantel–Cox test. Differences between tumor growth curves were determined by repeated measures of two-way ANOVA. Significance was set at a *P* value less than or equal to 0.05. For all figures, *$P < 0.05$, **$P < 0.01$, ***$P < 0.001$. Unless noted in the

figure legend, all data are shown as the mean ± SEM. Generally, all experiments were carried out with $n \geq 3$ biological replicates.

## Data availability

All data associated with this study are present in the paper, and original data have been uploaded. scRNA-seq and TCR-seq were deposited in GEO database (accession no. GSE243535).

The source data of this paper are collected in the following database record: biostudies:S-SCDT-10_1038-S44321-024-00184-1.

## Peer review information

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

## Acknowledgements

This work was supported by grants from the State Key Program of the National Natural Science Foundation of China (81930080), the Youth Fund of the National Natural Science Foundation of China (32300762), Science and Technology Plan Project of Huai'an (HAB2024023), the Key Research and the Nature Science Foundation of Jiangsu Province (BK20210027), and the Jiangsu Innovative and Entrepreneurial Talent Program (JSSCBS20211488).

## Author contributions

**Yuncheng Bei**: Conceptualization; Data curation; Funding acquisition; Investigation; Writing—original draft. **Ying Huang**: Data curation; Software. **Nandie Wu**: Resources; Software; Investigation. **Yishan Li**: Resources; Investigation; Methodology. **Ruihan Xu**: Formal analysis; Validation. **Baorui Liu**: Supervision; Project administration; Writing—review and editing. **Rutian Li**: Conceptualization; Supervision; Funding acquisition; Writing—review and editing.

Source data underlying figure panels in this paper may have individual authorship assigned. Where available, figure panel/source data authorship is listed in the following database record: biostudies:S-SCDT-10_1038-S44321-024-00184-1.

## Disclosure and competing interests statement

The authors declare no competing interests.

