## [Peer Review File · EMBO Molecular Medicine]

A therapeutic regimen using neoantigen-specific TCR-T cells for HLA-A*2402-positive solid tumors

Yuncheng Bei, Ying Huang, Nandie Wu, Yishan Li, Ruihan Xu, Baorui Liu, and Rutian Li

Corresponding authors: Rutian Li (rutianli@nju.edu.cn) , Baorui Liu (baoruiliu@nju.edu.cn)

Review Timeline:

Submission Date:	12th Mar 24
Editorial Decision:	17th Apr 24
Revision Received:	5th Sep 24
Editorial Decision:	4th Oct 24
Revision Received:	20th Nov 24
Accepted:	2nd Dec 24

Editor: Poonam Bheda

Transaction Report:

17th Apr 2024

Dear Prof. Li,

Thank you for the submission of your manuscript to EMBO Molecular Medicine. We have now received feedback from the two reviewers who agreed to evaluate your manuscript. As you will see from the reports below, the referees acknowledge the interest of the study and are overall supportive of your work; however they also comment on multiple aspects of the manuscript that should be strengthened in a revision.

In particular, both reviewers mentioned the need to generally improve the defining characteristics of the various immune cell types, i.e. employ additional marker expression to better cluster/define cell types. In addition, both suggested additional information/validation on the nanoparticle delivery system and its specificity. Reviewer 2 is concerned by the safety of the approach and encourages you to show the safety profile of the neoantigen peptide in a mouse that expresses humanized HLA with additional in vitro testing to evaluate the possible cross-reactivity. In general these concerns appear addressable in a reasonable timeframe.

Reviewer 1 comments that addition of cancer patient samples to show whether the tumor epitope is immunogenic in vivo would significantly improve the manuscript. While editorially we agree that this would significantly enhance the manuscript, we are also aware that addition of this type of data is not trivial if you do not already have access to and consent for a cohort of patient samples ready to test. Therefore, we would suggest that you may address this point if you have access and consent for these samples, but it will not be a requirement for a successful revision.

Addressing the reviewers' concerns in full in a point-by-point response will be necessary for further considering the manuscript in our journal, and acceptance of the manuscript will entail a second round of review. EMBO Molecular Medicine encourages a single round of revision only and therefore, acceptance or rejection of the manuscript will depend on the completeness of your responses included in the next, final version of the manuscript. For this reason, and to save you from any frustrations in the end, I would strongly advise against returning an incomplete revision. If you would like to discuss further the points raised by the referees, I am available to do so via email or video. Let me know if you are interested in this option.

We are expecting your revised manuscript within three months, if you anticipate any delay, please contact us. When submitting your revised manuscript, please carefully review the instructions that follow below. We perform an initial quality control of all revised manuscripts before re-review; failure to include requested items will delay the evaluation of your revision.

We require:

- 1) A .docx formatted version of the manuscript text (including legends for main figures, EV figures and tables). Please make sure that the changes are highlighted to be clearly visible.
- 2) Individual production quality figure files as .eps, .tif, .jpg (one file per figure). For guidance, download the 'Figure Guide PDF' (<https://www.embopress.org/page/journal/17574684/authorguide#figureformat>).
- 3) At EMBO Press we ask authors to provide source data for the main figures. Our source data coordinator will contact you to discuss which figure panels we would need source data for and will also provide you with helpful tips on how to upload and organize the files.
- 4) A .docx formatted letter INCLUDING the reviewers' reports and your detailed point-by-point responses to their comments. As part of the EMBO Press transparent editorial process, the point-by-point response is part of the Review Process File (RPF), which will be published alongside your paper.
- 5) A complete author checklist, which you can download from our author guidelines (<https://www.embopress.org/page/journal/17574684/authorguide#submissionofrevisions>). Please insert information in the checklist that is also reflected in the manuscript. The completed author checklist will also be part of the RPF.
- 6) Please note that all corresponding authors are required to supply an ORCID ID for their name upon submission of a revised manuscript.
- 7) It is mandatory to include a 'Data Availability' section after the Materials and Methods. Before submitting your revision, primary datasets produced in this study need to be deposited in an appropriate public database, and the accession numbers and database listed under 'Data Availability'. Please remember to provide a reviewer password if the datasets are not yet public (see <https://www.embopress.org/page/journal/17574684/authorguide#dataavailability>).

In case you have no data that requires deposition in a public database, please state so in this section. Note that the Data

Availability Section is restricted to new primary data that are part of this study.
This study includes no data deposited in external repositories.

8) For data quantification: please specify the name of the statistical test used to generate error bars and P values, the number (n) of independent experiments (specify technical or biological replicates) underlying each data point and the test used to calculate p-values in each figure legend. The figure legends should contain a basic description of n, P and the test applied. Graphs must include a description of the bars and the error bars (s.d., s.e.m.). Please provide exact p values.

13) Author contributions: CRedit has replaced the traditional author contributions section because it offers a systematic machine readable author contributions format that allows for more effective research assessment. Please remove the Authors Contributions from the manuscript and use the free text boxes beneath each contributing author's name in our system to add specific details on the author's contribution. More information is available in our guide to authors.

Please also suggest a striking image or visual abstract to illustrate your article as a PNG file 550 px wide x 300-600 px high. Share synopsis text and image, as well as eTOC:

Please note that these would be the final versions and changes during proofing are usually not allowed

16) As part of the EMBO Publications transparent editorial process initiative (see our policy here: https://www.embopress.org/transparent-process#Review_Process), EMBO Molecular Medicine will publish online a Peer Review File (PRF) to accompany accepted manuscripts.

In the event of acceptance, this file will be published in conjunction with your paper and will include the anonymous referee reports, your point-by-point response and all pertinent correspondence relating to the manuscript. Let us know whether you agree with the publication of the PRF and as here, if you want to remove or not any figures from it prior to publication. Please note that the Authors checklist will be published at the end of the RPF.

I look forward to receiving your revised manuscript.

Yours sincerely,

Poonam Bheda

Poonam Bheda, PhD
Scientific Editor
EMBO Molecular Medicine

***** Reviewer's comments *****

Referee #1 (Comments on Novelty/Model System for Author):

For specific comments, see remarks to the author.

Referee #1 (Remarks for Author):

This manuscript describes the identification and functional characterization (in vitro and in vivo) of a T cell receptor specific for a peptide that is generated through sarcoma-specific gene fusion. Overall, the work is fairly mature and relevant, since gene fusion-derived epitopes are not as widely reported as other neoantigens, but hold great promise for broader applicability of cancer T cell therapies. Of note, here a single T cell receptor against a single epitope was followed up due to its promising potential. This TCR was derived from the naïve (antigen-inexperienced) repertoire of a healthy donor. A frequent shortcoming in studies like the one at hand is that it is unclear whether the tumor epitope is immunogenic in vivo "in nature" (i.e. in cancer patients). If the authors were able to show such natural in vivo immunogenicity, the manuscript would significantly gain in quality. In addition, individual sections of the manuscript need major improvement, as outlined below.

Fig. 2: It is unclear whether only clusters 1 and 2 are CD8 T cell clusters or not. For all cells, there should be a definite cluster assignment using information such as, e.g. CD3, TRAC or TRBC expression in order to first define which cells belong to CD8 ab TCR T cells. It should then be better defined what the difference between cluster 1 and 2 is, e.g. by DEG analysis (volcano plot). Given the fact that TCR-1 T cells stand out in terms of e.g. GNLY expression, it is further unclear whether TCR-1 was the only TCR that reacted to Pep4 and the other TCRs (TCR-2 to TCR-7) are just bystander-expanded, or whether they are also antigen-reactive. From Fig. S4B and Fig. S4F, it only later becomes apparent that TCR-1 seems to be the only antigen-reactive TCR identified. Instead of the plots currently shown in Fig. 2F-G, it would be more clear if e.g. reactivity scores or gene expression for more classical candidate genes indicating T cell activation were shown for the 7 TCRs side-by-side.

Fig. 4: The immunophenotyping is insufficient. Merely CD137 and TEM differentiation are being investigated, and the data are not entirely convincing. Differences in CD137 are very mild in the primary data. The CD8 gating ("gate 1" in Fig. 4D) contains two sub-populations, neither of which look like a typical CD8 population. In the tumor, this may be due to CD8 downregulation, but appears to be more likely a technical artefact. In terms of CD45RO vs. CD62L, it is surprising that there is no gradual transition but there appear to be two defined populations only. The authors need to a) clarify the mentioned technical issues, b) investigate more markers via flow cytometry, including more activation markers, a full assessment of all differentiation stages (i.e. at least naïve, TCM, TEM, TEF) and an analysis of co-inhibitory markers (e.g. PD-1, TIM-3, LAG-3). This part would also be improved if additional phenotyping by scRNAseq was performed.

Fig. 5: It needs to be explained in more detail in the text how tumor targeting via the nanoparticles actually works (i.e. the scheme of Fig. 5A needs to be explained in terms that are understandable to a reader not familiar with the technique). In particular, it remains unclear how tumor specificity is reached. For the analyses shown in Fig. 5E-F, negative controls need to be shown.

Fig. 6: This section would benefit from side-by-side comparisons of protection from tumors endogenously expressing the Pep-4 peptide vs. tumors missing it, which are then targeted by the nanoparticle. Ideally, the Pep-4 vs. delta-Pep-4 tumors from Fig. 3 would be used for this. The reference to Fig. 6F is missing in the text.

In the discussion section referring to phenotype-based identification of high-avidity tumor-specific TCRs, the work by Purcarea et al. (PMID: 35960818) should be mentioned. Also, the work by Lucca et al. (PMID: 33651881) and Meng et al. (PMID: 37967201)

should be referenced with regards to phenotype-based identification of tumor-specific TCRs (irrespective of avidity). In the discussion, a limitations section also needs to be incorporated, highlighting the shortcomings of this study. Language-wise, throughout the manuscript, „conservative epitope" should be changed into "conserved epitope". Self-referring appraisal like "innovatively" in abstract and introduction should be omitted. The last sentence of the introduction is weird ("Cooperatively" should be deleted). "Originally" in line 500 and 644 should also be omitted. There are a few other text passages that need refinement in terms of language. The figures need a visual work-up. For example, the axis labels on the flow cytometry primary data plots are redundant and of low visual quality.

Referee #2 (Comments on Novelty/Model System for Author):

The major concern is that the model used does not express the human HLA that matches the TCR restriction, therefore even if the neoag is introduced via the gelatinase-responsive nanoparticles it is not possible to evaluate the impact on healthy tissues.

Referee #2 (Remarks for Author):

The manuscript "A universal therapeutic regimen using neoantigen-specific TCR-engineered T cells for solid tumors" describes a possible therapeutic approach where tumor cells are artificially loaded with a specific neo-antigen to allow recognition from specific TCR-T cells.

The manuscript is overall well written but there are some major points that need to be addressed:

- 1) The title is misleading because the study is limited to a specific type of tumor and a single neoantigen with a single HLA restriction, therefore not relatable to multiple solid tumors and definitely not "universal". The authors should change the title and highlight these limitations in the discussion. Alternatively they should provide more data for additional solid tumors, neoantigens and HLA restrictions.
- 2) Another major issue is related to the safety of the approach. The in vivo model is a mouse that does not express human HLA, therefore it is expected that the specific TCR will be unable to recognize healthy tissue, but this is not a proof of low toxicity. The authors should show the safety profile in a mouse that express HLA-A24 in all tissues and not only in a xeno model. Moreover additional in vitro testing should be performed to evaluate the possible cross-reactivity of the TCR to similar peptides (an alanin-substitution experiment could be a good start).

After solving these major issues there are some minor points that should be considered too:

- 1) Gelatinase-responsive nanoparticle should be loaded with additional possible targets that show similarity with the specific target to have a more accurate proof of specificity in vivo.
- 2) T cells transduced with different TCRs should be used as control instead of UTD to take into account the possible increased level of cytotoxicity due to the cell activation.
- 3) The data on the TCR-T cell infiltration should not be based only on 4-1BB expression, possible markers could be a specific dextramer or the labelling of TCR-T cells.
- 4) In the introduction, line 59 the sentence "the sufficient safety and effectiveness remain to be well documented due to the deficiency of their specificity." Should be referenced.

Dear reviewers,

Thank you very much for your grateful, positive, constructive evaluations of our work. We have addressed the major points from reviewers, and responded to reviewers' comments point by point, as shown by the following. We feel that the inclusion of these additional data greatly strengthens our study and supports our hypothesis.

For Referee #1 (Comments on Novelty/Model System for Author):

This manuscript describes the identification and functional characterization (in vitro and in vivo) of a T cell receptor specific for a peptide that is generated through sarcoma-specific gene fusion. Overall, the work is fairly mature and relevant, since gene fusion-derived epitopes are not as widely reported as other neoantigens, but hold great promise for broader applicability of cancer T cell therapies. Of note, here a single T cell receptor against a single epitope was followed up due to its promising potential. This TCR was derived from the naïve (antigen-inexperienced) repertoire of a healthy donor. A frequent shortcoming in studies like the one at hand is that it is unclear whether the tumor epitope is immunogenic in vivo "in nature" (i.e. in cancer patients). If the authors were able to show such natural in vivo immunogenicity, the manuscript would significantly gain in quality. In addition, individual sections of the manuscript need major improvement, as outlined below.

Response: Thanks for this suggestion. We are in complete agreement with your viewpoint that figuring out whether this fusion-mutation tumor epitope is immunogenic in vivo is very important for further clinical investigation. Indeed, Satoshi Kawaguchi and colleagues previously proved that SYT-SSX gene-derived peptides are of great immunogenicity both in vitro and in patients with synovial sarcoma (PMID: 15647119). More importantly, in this phase I clinical trial, vaccinations with a 9-mer peptide, similar to the identified peptide in this study, successfully suppressed tumor progression with no serious adverse effects. This phase I vaccination trial gives us a hint that SYT-SSX fusion-derived peptides are immunogenic in patients. However, we still believe that it is necessary to perform clinical trials to investigate the efficacy of this TCR-T-based cooperative therapeutic modality. Thus, a related investigator-initiated clinical trial (IIT) is in preparation. We are willing to present detailed information in our next work.

Fig. 2: It is unclear whether only clusters 1 and 2 are CD8 T cell clusters or not. For all cells, there should be a definite cluster assignment using information such as e.g. CD3, TRAC, or TRBC expression to first define which cells belong to CD8 ab TCR T cells. It should then be better defined what the difference between Cluster 1 and Cluster 2 is, e.g. by DEG analysis (volcano plot). Given the fact that TCR-1 T cells stand out in terms of e.g. GNLY expression, it is further unclear whether TCR-1 was the only TCR that reacted to Pep4 and the other TCRs (TCR-2 to TCR-7) are just a bystander-expanded, or whether they are also antigen-reactive. From Fig. S4B

and Fig. S4F, it only later becomes apparent that TCR-1 seems to be the only antigen-reactive TCR identified. Instead of the plots currently shown in Fig. 2F-G, it would be more clear if e.g. reactivity scores or gene expression for more classical candidate genes indicating T cell activation were shown for the 7 TCRs side-by-side.

Response: Thanks for this suggestion. All cells were filtered by CD3 expression (Fig. S3C revised version). The cluster 1 and 2 dominantly belonged to CD8+ T-cells. Different expression genes were shown in Fig. 2D. Comparing with cluster 2, classical genes, such as IFN γ , IL2RA, and MYB were elevated in cluster 1 indicating T-cell activation. Additionally, we also noticed that Mitosis-related genes (MKI67, TK1, and STMN1) were highly upregulated (Fig. 2C-D). The GO (gene ontology) analysis strengthened the conclusion that T-cells in cluster 1 were potentially antigen-responsive (Fig. 2E). Although, T-cells expressing Tcr-1 belonged to cluster 1, and T-cell activation-related genes were highly upregulated compared with T-cells expressing other TCRs, it was still inadequate to conclude that Tcr-1 was Pep-4-responsive TCR. Target-specific killing assay was necessary for further determination (Fig S4B). As a result, from Tcr-2 to Tcr-7 were seemingly bystander-expanded.

Fig. 4: The immunophenotyping is insufficient. Merely CD137 and TEM differentiation are being investigated, and the data are not entirely convincing. Differences in CD137 are very mild in the primary data. The CD8 gating ("gate 1" in Fig. 4D) contains two sub-populations, neither of which look like a typical CD8 population. In the tumor, this may be due to CD8 downregulation but appears to be more likely a technical artifact. In terms of CD45RO vs. CD62L, it is surprising that there is no gradual transition but there appear to be two defined populations only. The authors need to a) clarify the mentioned technical issues, b) investigate more markers via flow cytometry, including more activation markers, a full assessment of all differentiation stages (i.e. at least naïve, TCM, TEM, TEF), and analysis of co-inhibitory markers (e.g. PD-1, TIM-3, LAG-3). This part would also improved if additional phenotyping by scRNAseq was performed.

Response: Thanks for these suggestions. a) we performed the similar test to verify the Naïve, TCM and TEM by using fluorescent antibodies newly purchased from BD (CD3-BV510, Cat no. 564713; CD4-APC-Cy7, Cat no. 561839; CD8-BV605, Cat no. 564116; CD45RO-PE, Cat no. 561889; CD62L-BV421, Cat no. 563862). Consistent with our prediction, upon exposure to the Pep-4/HLA-A*2402 complex, Tcr-T1 cells possessed a high proportion of Tem subpopulations both in tumor tissues and spleen tissues. By using the new antibody, we could see a gradual transition as the reviewer mentioned. Thus, the decreased antibody quality might make a dominant contribution to the atypical image. b) we performed further flow cytometric analysis to investigate co-inhibitory marker expression. As shown in Fig. 4E-F, PD-1, and TIM-3 were highly expressed in Tcr-T1 cells upon recognition of the Pep-4/HLA-A*2402 complex in tumor tissues. To our knowledge, PD-1 and TIM-3 were elevated after T-cell activation as negative feedback regulators. These data not only proved that Tcr-T1 cells were activated dependent on Pep-4 neoantigen

expression but also provided hints that combining immune checkpoint inhibitors might facilitate Tcr-T1 therapy efficacy. We agree that scRNA-seq could provide more improvement. And we think it is more suitable for phenotypic analysis in the next clinical investigation in the future.

Fig. 5: It needs to be explained in more detail in the text how tumor targeting via the nanoparticles works (i.e. the scheme of Fig. 5A needs to be explained in terms that are understandable to a reader not familiar with the technique). In particular, it remains unclear how tumor specificity is reached. For the analyses shown in Fig. 5E-F, negative controls need to be shown.

Response: Detailed information about how tumor antigens were transferred to tumor tissues was provided “This therapeutic strategy contained two steps. In the first step, tumor-specific neoantigens were delivered to tumor tissues using gelatinase-responsive nanoparticles. It is widely reported that matrix metalloproteinase (MMP) 2/9 was enriched in various solid tumor tissues, playing a key role in cancer invasion and metastasis (Kessenbrock et al, 2010), which have been regarded as a favorable target for drug delivery (Zandieh et al, 2023). Thus, we aimed to load Pep-4 neo-peptide into the stimuli-responsive nanoparticles (mPEG-PVGLIG-PCL copolymer) as described (Li et al, 2013), which can accumulate in the tumor site via EPR effects. Upon tumor-specific MMP2/MMP9-mediated digestion, loaded Pep-4 neo-peptide was released in tumor tissues, competitively binding to tumor HLA. At the second step, Pep-4-specific TCR-T cells were adoptively transferred.” (highlighted in lines 502-512).

To our knowledge, MMP2/9 was specifically enriched in various solid tumors, exhibiting potentiality as the favorable target for stimuli-responsive nanoparticle establishment (PMID: 37567073). Indeed, in this study we observed that Pep-4-loaded mPEG-PVGLIG-PCL copolymer was enriched in tumor tissue (Fig. 5E). Upon MMP2/9 digestion, Pep-4 neopeptide was released and conjugated with HLA-A*2402 on tumor cells (Fig. 5D). More importantly, in-vivo test indicated that Pep-4 neopeptide was stable for at least 96 hours, while it could not be detected in other normal tissues (Fig. 5F). Probably, binding with HLA molecule enhanced stability of Pep-4. These data indicated that this cooperative therapeutic strategy has a favorable tumor-targeted specificity.

Previous data showed that Pep-2 could not bind to the HLA-A*2402 molecule. Thus, it has been used as a negative control in Fig. 5E-F.

Fig. 6: This section would benefit from side-by-side comparisons of protection from tumors endogenously expressing the Pep-4 peptide vs. tumors missing it, which are then targeted by the nanoparticle. Ideally, the Pep-4 vs. delta-Pep-4 tumors from Fig. 3 would be used for this. The reference to Fig. 6F is missing in the text.

Response: Thanks for this suggestion. To further investigate the specificity and efficacy of the therapeutic regimen, we performed the mouse model, in one mouse bears two types of tumors

side-by-side. As shown in Fig. 6F, consistent with our previous results, adoptive transferring Tcr-T1 could specifically attenuate HLA-A*2402-positive MKN-45 tumor growth, while having little effect on HLA-A*0201-positive MDA-MB-231 cells. These data indicated that Pep-4 NP-mediated tumor-targeted delivery of neo-peptide successfully provided targets for Tcr-T1 cells, and functioned to trigger the recognition, engagement, and final tumor elimination independent of tumor inherent mutation profiles, with favorable specificity and efficiency.

The reference to Fig. 6F (Fig. 6G in the revised version) is provided in line 561.

In the discussion section referring to phenotype-based identification of high-avidity tumor-specific TCRs, the work by Purcarea et al. (PMID: 35960818) should be mentioned. Also, the work by Lucca et al. (PMID: 33651881) and Meng et al. (PMID: 37967201) should be referenced with regard to phenotype-based identification of tumor-specific TCRs (irrespective of avidity). In the discussion, a limitations section also needs to be incorporated, highlighting the shortcomings of this study.

Response: PMID:35960818 reference was added (line 632). PMID: 33651881 references were added (lines 609-612). PMID: 33651881 references were added (line 638).

A limitation section was provided (lines 658-664, highlighted in the revised manuscript)

Language-wise, throughout the manuscript, „conservative epitope" should be changed into "conserved epitope". Self-referring appraisal like "innovatively" in abstract and introduction should be omitted. The last sentence of the introduction is weird ("Cooperatively" should be deleted). "Originally" in line 500 and 644 should also be omitted. There are a few other text passages that need refinement in terms of language.

Response: Sorry for these mistakes. The “conservative epitope” was replaced by the "conserved epitope”. The self-referring appraisal is omitted. The “cooperatively” in the last sentence of the introduction was deleted. "Originally" in lines 500 and 644 were also omitted. The language has been checked again.

The figures need a visual work-up. For example, the axis labels on the flow cytometry primary data plots are redundant and of low visual quality.

Response: The figures have been refined. The redundant axis labels were removed.

Referee #2 (Comments on Novelty/Model System for Author):

The major concern is that the model used does not expressed the human HLA that matches the

TCR restriction, therefore even if the neoag is introduced via the gelatinase-responsive nanoparticles it is not possible to evaluate the impact on healthy tissues.

Referee #2 (Remarks for Author):

The manuscript "A universal therapeutic regimen using neoantigen-specific TCR-engineered T cells for solid tumors" describes a possible therapeutic approach where tumor cells are artificially loaded with a specific neo-antigen to allow recognition from specific TCR-T cells.

The manuscript is overall well written but there are some major points that need to be addressed:

1) The title is misleading because the study is limited to a specific type of tumor and a single neoantigen with a single HLA restriction, therefore not relatable to multiple solid tumors and definitely not "universal". The authors should change the title and highlight these limitations in the discussion. Alternatively they should provide more data for additional solid tumors, neoantigens and HLA restrictions.

Response: We agree with the reviewer's opinion. We have changed the title and pointed out the limitation of this study in the discussion section (lines 686-692). To further investigate the specificity and efficacy of this cooperative therapeutic regimen, we performed the mouse model, in one mouse bears two types of tumors side-by-side. As shown in Fig. 6F, consistent with our previous results, adoptive transferring Tcr-T1 could specifically attenuate HLA-A*2402-positive MKN-45 tumor growth, while having little effect on HLA-A*0201-positive MDA-MB-231 cells after administration of Pep-4 NP. These data indicated that Pep-4 NP-mediated tumor-targeted delivery of neo-peptide successfully provided targets for Tcr-T1 cells, and functioned to trigger the recognition, engagement, and final tumor elimination independent of tumor inherent mutation profiles, with favorable specificity and efficiency.

In this study, we have tested the therapeutic efficacy both in vitro and in vivo on synovial sarcomas (sw982 cell line) and gastric cancer (MKN45). The cooperative therapeutic regimen showed remarkable functions in various mouse models, including the subcutaneous xenograft tumor model and peritoneal metastasis tumor model. These data indicated that the cooperative regimen might show great activity against various solid tumors, however in an HLA-restricted manner.

2) Another major issue is related to the safety of the approach. The in vivo model is a mouse that does not express human HLA, therefore it is expected that the specific TCR will be unable to recognize healthy tissue, but this is not a proof of low toxicity. The authors should show the safety profile in a mouse that express HLA-A24 in all tissues and not only in a xeno model. Moreover additional in vitro testing should be performed to evaluate the possible cross-reactivity of the TCR

to similar peptides (an alanin-substitution experiment could be a good start).

Response: We agree with this suggestion. To make this investigation, mice that express human HLA-A*2402 in all tissues are needed. We have contacted with GemPharmatech and found that the mice (BALB/cJGpt-b2m^{em1Cin(hHLA-A24.2,hB2m, H2-D1)}/Gpt, Strain No. T064364) were in prepare, but not for sale. More importantly, to investigate the side effects, these HLA-A*2402-expressing mice should be hybridized with immunodeficiency mice (NOD/ShiLtJGpt-Prkdc^{em26Cd52}-Il2rg^{em26Cd22}/Gpt) and screened furtherly to generated homozygous immunodeficiency mice expressing HLA-A*2402. Unfortunately, it is time-consuming. To avoid this potential on-target off-tumor side effect, we have previously investigated the tissue distribution of Pep-4 NP in the mouse model (Fig. 5E-G). Pep-4 NP was significantly enriched in tumor tissues (Fig. 5E), and stabilized at least for 48 hours (Fig. 5F). These data guided us to adaptively transfer Tcr-T1 cells in 24 hours after Pep-4 NP administration. Generally, the cooperative regimen showed remarkable anti-tumoral activity with favorable specificity and good tolerance. However, we agree with the reviewer's opinion that these data are still insufficient, and a clinical investigation is very necessary. A related investigator-initiated clinical trial (IIT) is in preparation. We are willing to present detailed information in our next work.

After solving these major issues there are some minor points that should be considered too:

1) Gelatinase-responsive nanoparticle should be loaded with additional possible targets that show similarity with the specific target to have a more accurate proof of specificity in vivo.

Response: To further investigate the specificity and efficacy of the therapeutic regimen, we performed the mouse model, that one mouse bears two types of tumors side-by-side. As shown in Fig. 6F, consistent with our previous results, adoptive transferring Tcr-T1 could specifically attenuate HLA-A*2402-positive MKN-45 tumor growth, while having little effect on HLA-A*0201-positive MDA-MB-231 cells. These data indicated that Pep-4 NP-mediated tumor-targeted delivery of neo-peptide successfully provided targets for Tcr-T1 cells, and functioned to trigger the recognition, engagement, and final tumor elimination independent of tumor inherent mutation profiles, with favorable specificity and efficiency.

2) T cells transduced with different TCRs should be used as control instead of UTD to take into account the possible increased level of cytotoxicity due to the cell activation.

Response: To our knowledge, T cells express endogenous TCRs. UTD cells are composed of thousands of subclones of TCRs, which could be a good control. Additionally, the specificity of cytotoxicity could be well-defined by transducing target cells with neoantigen or not.

3) The data on the TCR-T cell infiltration should not be based only on 4-1BB expression, possible markers could be a specific dextramer or the labelling of TCR-T cells.

Response: In this study, an artificial RQR8 epitope (described in PMID: 24970931) was used as an identification and isolation marker of TCR-T cells in vitro. TCR-T cells were identified by using fluorescent-labeled antibodies targeting human CD markers in vivo, such as CD3, CD8, and CD4. CD137 (4-1BB) is used as a marker to indicate T-cell activation.

4) In the introduction, line 59 the sentence "the sufficient safety and effectiveness remain to be well documented due to the deficiency of their specificity." Should be referenced.

Response: the reference was added.

4th Oct 2024

Dear Prof. Li,

Thank you for the submission of your revised manuscript to EMBO Molecular Medicine. Your manuscript has now been re-reviewed by both of the original reviewers. Based on their advice (included below), I am pleased to inform you that we will be able to accept your manuscript pending the following final amendments and appropriate response to reviewers:

- 1) Please check the "Author Checklist" carefully and complete all relevant questions. While you have indicated that cell lines were tested for mycoplasma and authenticated in the manuscript, this section in the author checklist is currently marked as 'not applicable'. Please update accordingly.
- 2) Please ensure that co-corresponding author Baorui Liu has linked their ORCID ID to their profile in our system as this is required for corresponding authors.
- 3) Please rephrase both Introduction and Discussion, as we identified substantial overlap with published literature. Please be aware that even if the content is largely from your own papers, whole sentences and phrases should not be copied.
- 4) Please rename the 'Data and materials availability' statement to 'Data availability'. Please also remove the reviewer token for the sequencing data and ensure that these datasets are now released and publicly available. Please be aware that all deposited datasets should be freely accessible prior to publication.
- 5) Please rename "Authors disclosure" to "Disclosure and competing interests statement". We updated our journal's competing interests policy in January 2022 and request authors to consider both actual and perceived competing interests. Please review the policy <https://www.embopress.org/competing-interests> and update your competing interests if necessary.
- 6) Author contributions: Please remove it from the manuscript and specify author contributions in our submission system. CRediT has replaced the traditional author contributions section because it offers a systematic machine-readable author contributions format that allows for more effective research assessment. You are encouraged to use the free text boxes beneath each contributing author's name to add specific details on the author's contribution. More information is available in our guide to authors:
<https://www.embopress.org/page/journal/17574684/authorguide#authorshipguidelines>
- 7) In the Methods, please take care of the following:
 - Please rename this section from 'Materials and methods' to 'Methods'
 - Animals: Please ensure that gender and age of the animals involved in experiments is reported in the Methods.
 - Antibodies: information on the TIM3 antibody used for flow cytometry appears to be missing in the Methods. Please double check this and add the information accordingly.
- 8) All materials and methods need to be described in the main text using our 'Structured Methods' format, which is required for all research articles. According to this format, the Methods section includes a Reagents and Tools Table (listing key reagents, experimental models, software and relevant equipment and including their sources and relevant identifiers) followed by a Methods and Protocols section describing the methods using a step-by-step protocol format. The aim is to facilitate adoption of the methodologies across labs. More information on how to adhere to this format as well as a downloadable template (.docx) for the Reagents and Tools Table can be found in our author guidelines:
<https://www.embopress.org/page/journal/17574684/authorguide#structuredmethods>

- 9) Please place individual sections of the manuscript in the following order: Title page - Abstract & Keywords - Introduction - Results - Discussion - Methods - Data Availability - Acknowledgements - Disclosure and Competing Interests Statement - The Paper Explained - References - Figure Legends - Expanded View Figure Legends.
- 10) For the figures and figure legends, please take care of the following:
 - Please make sure to update the callouts of all figures and tables in the main manuscript text, which should be numbered according to the order in which they appear. Currently Table S2 is called out before S1; a callout is missing for Table and for Fig 3C.
 - Please note that we require exact p-values to be reported. Currently exact p-values are not provided in the legends of figures 1e-h; 3c-f, h-i; 4a, c-f; 5d, f; 6a, c-d, g.
 - Please indicate the statistical test used for data analysis in the legends of figures 1e; 2d-e; 3i; 4d.
 - Please note that information related to n is missing in the legends of figures 1e-h; 2i-j; 3e-f; 4e-f; 6c, e.
 - Although 'n' is provided, please describe the nature of entity for 'n' in the legends of figures 4c-d; 5d; 6a.
 - Please note that the scale bar needs to be defined for figures 1h; 5g.
- 11) The supplementary information should be renamed "Appendix", uploaded in PDF format, and a table of contents should be added to the first page. The supplementary methods should be removed from the file and merged with the methods in the manuscript text. The suppl. figures and legends should be renamed "Appendix Figure S1" etc. Supplementary tables S1-3 should be added to the appendix in PDF format and renamed "Appendix Table S1" etc.
- 12) Synopsis:
 - Synopsis image: Please edit your synopsis image to ensure that it fits into our requested dimensions of 550 pixels wide x (250-400) pixels high. Although we can resize the image for you, currently when resized to 550 pixels wide, the image is too high.

- Synopsis text: Please provide a short standfirst (maximum of 300 characters, including space), limit the bullet points to max. 5 and upload it as a separate .doc file. Please write the bullet points to summarise the key NEW findings. They should be designed to be complementary to the abstract - i.e. not repeat the same text. We encourage inclusion of key acronyms and quantitative information (maximum of 30 words / bullet point). Please use the passive voice.

13) The Paper Explained: Please provide "The Paper Explained" and add it to the main manuscript text. Please check "Author Guidelines" for more information. <https://www.embopress.org/page/journal/17574684/authorguide#researcharticleguide>

14) As part of the EMBO Publications transparent editorial process initiative (see our policy here:

https://www.embopress.org/transparent-process#Review_Process), EMBO Molecular Medicine will publish online a Peer Review File (PRF) to accompany accepted manuscripts. This file will be published in conjunction with your paper and will include the anonymous referee reports, your point-by-point response and all pertinent correspondence relating to the manuscript. Let us know whether you agree with the publication of the PRF and as here, if you want to remove or not any figures from it prior to publication. Please note that the Authors checklist will be published at the end of the PRF.

15) Please provide a point-by-point letter INCLUDING my comments as well as the reviewer's reports and your detailed responses (as Word file).

I look forward to reading a new revised version of your manuscript as soon as possible.

Yours sincerely,

Poonam Bheda

Poonam Bheda, PhD
Scientific Editor
EMBO Molecular Medicine

***** Reviewer's comments *****

Referee #1 (Comments on Novelty/Model System for Author):

-

Referee #1 (Remarks for Author):

The authors have addressed most of my points to a somewhat satisfactory extent. There is a couple of things still to be done:
Line 22: The first sentence of the abstract twice entails the word preferentially. Please omit the word in the second instance.

Line 24: "we first identified a paired a/b TCR repertoire" -> A single TCR, not a repertoire, was identified. The word repertoire is misleading.

Line 28: Omit the word "furtherly" since "extend" already conveys this meaning.

The language of the entire manuscript still needs to be revised thoroughly to bring it on native-speaker level and thus easier to read.

Fig. 3D: The data points belonging to the 4 conditions are not distinguishable in the figure. It can only be guessed that the dark red condition is the only one with a decrease in cell viability.

Fig. 4: The immunophenotyping is now better presented, but I would suggest to be more careful with the interpretation. I see the stronger TEM fraction (which is effectively only due to lower levels of CD62L) in responsive TCR-T1 only being due to T cell activation happening. The same applies to heightened PD-1 and TIM-3 expression. Since the authors are observing these phenotypes in a somewhat artificial mouse model, I would restrict the interpretation to TCR-T1 cells effectively recognizing antigen in vivo, but omit speculation that protection is mediated by TEM cells or that TIM-3 blocking could help, because it is insinuated that both would apply for humans.

Fig. 5: The explanation on the nanoparticle system is still insufficient. The figure legend should contain far more detail, for example by explaining all abbreviations. What does "gelatinase-responsive" (line 539) mean and how is this connected to MMP 2/9 expression? It should be explained that MMP 2 and 9 are, in fact, gelatinases. This would make it a bit clearer that the whole concept is that antigen-loaded nanoparticles are digested (and then release the antigen) at the tumor site in a presumably site-specific manner. What does "stimuli-responsive" (line 543) mean? What does "EPR" (line 544) stand for? I assume it means "enhanced permeability and retention".

Referee #2 (Comments on Novelty/Model System for Author):

The authors have highlighted the limitations so it is more appropriate now

Referee #2 (Remarks for Author):

Thank you for addressing my comments

***** Reviewer's comments *****

Referee #1 (Comments on Novelty/Model System for Author):

-

Referee #1 (Remarks for Author):

The authors have addressed most of my points to a somewhat satisfactory extent. There is a couple of things still to be done:

Line 22: The first sentence of the abstract twice entails the word preferentially. Please omit the word in the second instance.

Response: The repetitive “preferentially” has been deleted.

Line 24: "we first identified a paired a/b TCR repertoire" -> A single TCR, not a repertoire, was identified. The word repertoire is misleading.

Response: This sentence has been corrected.

Line 28: Omit the word "furtherly" since "extend" already conveys this meaning.

Response: This sentence has been corrected

The language of the entire manuscript still needs to be revised thoroughly to bring it on native-speaker level and thus easier to read.

Fig. 3D: The data points belonging to the 4 conditions are not distinguishable in the figure. It can only be guessed that the dark red condition is the only one with a decrease in cell viability.

Response: Only one group has a significant decrease in cell viability, while other three groups have nearly the same cell viability. To better distinguish them, the exact p-value and “***” with the same color were provided.

Fig. 4: The immunophenotyping is now better presented, but I would suggest to be more careful with the interpretation. I see the stronger TEM fraction (which is effectively only due to lower levels of CD62L) in responsive TCR-T1 only being due to T cell activation happening. The same applies to heightened PD-1 and TIM-3 expression. Since the authors are observing these phenotypes in a somewhat artificial mouse model, I would restrict the interpretation to TCR-T1 cells effectively recognizing antigen in vivo, but omit speculation that protection is mediated by TEM cells or that TIM-3 blocking could help, because it is insinuated that both would apply for humans.

Response: Thanks for this suggestion. The classical T-cell activation markers, PD-1 and TIM-3, are also reported to negatively regulate T-cell activation. Thus, combined application of PD-1 or TIM-3 blocking antibody might promote anti-tumoral activity. This suggestion also provides a hint that blocking PD-1 or TIM-3 might enhance efficacy of adoptive transfer of TCR-T1 cells.

Fig. 5: The explanation on the nanoparticle system is still insufficient. The figure legend should contain far more detail, for example by explaining all abbreviations. What does "gelatinase-responsive" (line 539) mean and how is this connected to MMP 2/9 expression? It should be explained that MMP 2 and 9 are, in fact, gelatinases. This would make it a bit clearer that the whole concept is that antigen-loaded nanoparticles are digested (and then release the antigen) at the tumor site in a presumably site-specific manner. What does "stimuli-responsive" (line 543) mean? What does "EPR" (line 544) stand for? I assume it means "enhanced permeability and retention".

Response: To make it easily understand, "gelatinase-responsive" was replaced by "stimuli-responsive". The explanation of the nanoparticle system is provided in Results section "6. A cooperative therapeutic strategy for antigen-negative tumors.". And the abbreviations were provided in legend.

Referee #2 (Comments on Novelty/Model System for Author):

The authors have highlighted the limitations so it is more appropriate now

Referee #2 (Remarks for Author):

Thank you for addressing my comments

2nd Dec 2024

Dear Prof. Li,

We are pleased to inform you that your manuscript is accepted for publication and is now being sent to our publisher to be included in the next available issue of EMBO Molecular Medicine.

Yours sincerely,

Poonam Bheda, PhD
Scientific Editor
EMBO Molecular Medicine
